# How did the characteristics of the growing season change during the past 100 years at a steep river basin in Japan?

Nagai Shin[1,2]*, Taku M. Saitoh[2], Kenlo Nishida Nasahara[3]

**1** Earth Surface System Research Center, Research Institute for Global Change, Japan Agency for Marine-Earth Science and Technology (JAMSTEC), Kanazawa-ku, Yokohama, Japan, **2** River Basin Research Center, Gifu University, Gifu, Japan, **3** Faculty of Life and Environmental Sciences, University of Tsukuba, Tsukuba, Ibaraki, Japan

* nagais@jamstec.go.jp

**Data Availability Statement:** All data is publicly available from each Web site. Phenology images: http://www.pheno-eye.org https://koitoyaki.com/livecam/ (in Japanese, not archived) https://www.takayama-dp.com/live/ (in Japanese, not archived)

## Abstract

The effects of climate change on plant phenological events such as flowering, leaf flush, and leaf fall may be greater in steep river basins than at the horizontal scale of countries and continents. This possibility is due to the effect of temperature on plant phenology and the difference between vertical and horizontal gradients in temperature sensitivities. We calculated the dates of the start (SGS) and end of the growing season (EGS) in a steep river basin located in a mountainous region of central Japan over a century timescale by using a degree-day phenological model based on long-term, continuous, in situ observations. We assessed the generality and representativeness of the modelled SGS and EGS dates by using phenological events, live camera images taken at multiple points in the basin, and satellite observations made at a fine spatial resolution. The sensitivity of the modelled SGS and EGS dates to elevation changed from 3.29 days (100 m)$^{-1}$ (−5.48 days ˚C$^{-1}$) and −2.89 days (100 m)$^{-1}$ (4.81 days ˚C$^{-1}$), respectively, in 1900 to 2.85 days (100 m)$^{-1}$ (−4.75 days ˚C$^{-1}$) and −2.84 day (100 m)$^{-1}$ (4.73 day ˚C$^{-1}$) in 2019. The long-term trend of the sensitivity of the modelled SGS date to elevation was −0.0037 day year$^{-1}$ per 100 m, but the analogous trend in the case of the modelled EGS date was not significant. Despite the need for further studies to improve the generality and representativeness of the model, the development of degree-day phenology models in multiple, steep river basins will deepen our ecological understanding of the sensitivity of plant phenology to climate change.

## 1. Introduction

In temperate and subarctic climate zones, plant phenological events such as flowering, leaf flushing, leaf colouring, and leaf fall show spatiotemporal characteristics along latitudinal, longitudinal, and elevational gradients (e.g., [1–10]). In particular, the timing of the start (SGS) and end of the growing season (EGS) are more sensitive to elevational gradients than to latitudinal gradients [8]. This difference in sensitivity is caused by the difference in the sensitivity of temperature to gradients associated with elevation versus gradients associated with latitude

Meteorological data: https://www.data.jma.go.jp/gmd/risk/obsdl/index.php (in Japanese) Satellite data: https://creodias.eu.

**Funding:** KAKENHI grant (19H03301) from the Japan Society for the Promotion of Science https://www.jsps.go.jp/english/e-grants/index.html.

**Competing interests:** The authors have declared that no competing interests exist.

and longitude, because temperature affects the SGS and EGS dates (e.g., [11,12]). For instance, the sensitivities of temperature to elevation and latitude are 154 m $˚C^{-1}$ and about 122 km $˚C^{-1}$, respectively [13]. The sensitivity to elevation suggests that the effect of climate change on plant phenology may be apparent in a steep river basin within a distance of no more than a few tens of kilometres, much shorter than the latitudinal scales of countries and continents (hundreds to thousands of kilometres). For instance, a 3˚C increase in annual average temperature would shift the climate by an amount corresponding to a decrease in elevation of about 460 m. For this reason, examination of the relationship between SGS and EGS dates versus climate change along an elevational gradient is a useful way to estimate the spatiotemporal changes of SGS and EGS dates in a warmer climate.

Previous studies examined the spatiotemporal distribution of the timing of leaf flushing, leaf colouring, and leaf fall along an elevational gradient using data from in situ observations [14–17], phenological images [7,18,19], herbarium records [10], phenological information published on the Internet [9], data from satellite observations [6,12,20–23], and modelling [24,25]. The scales of the targeted regions have been national (Germany [15], Slovakia [17], and China [12,22,24]), continental (Europe [14] and North America [7,10]), and global [6]. In contrast, previous studies that focused on the scale of river basins (10–100 km) involved relatively limited spatiotemporal scales (a mountainous region in central Japan and 12 years [26]; a mountainous region in central Japan and 6 years [27]; a mountainous region in central Japan and 3 years [19]; a mountainous region in central Japan and 1 year [28,29]; the Pyrenees in France and 3 years [20]; the Pyrenees in France and 2 years [11]; and the Alps in Germany [18,30]). Four considerations may account for these relatively small spatiotemporal scales. First, the steep basins that are found in the mountainous region of central Japan (the Japan Alps) and the Alps, in which there are large differences in elevation within a short horizontal distance (1000–2000 m in elevation versus a horizontal distance of 10–50 km), are not widely distributed in the world. Second, long-term, in situ observations at multiple points in a river basin are labour-intensive and expensive. Third, satellite sensors with a spatial resolution of 500–1000 m (MODIS [Moderate Resolution Imaging Spectroradiometer] sensors mounted on Terra and Aqua satellites and the VEGETATION sensor mounted on the SPOT [Satellite pour l'Observation de la Terre] satellite) cannot accurately detect the spatiotemporal variability of the SGS and EGS dates because of the heterogeneity of vegetation [31] and microtopography. Fourth, data obtained from these satellite observations have been limited to only the past 20 years.

Taking account of these issues will require carrying out the following tasks in an integrated manner: (1) the long-term SGS and EGS dates should be calculated using a degree-day phenology model that is based on long-term, continuous biometeorological observations [1,16,32–35]; (2) the accuracy of the modelled SGS and EGS dates should be assessed by using long-term, continuous phenological images taken at multiple sites [31,36]; and (3) the spatiotemporal distribution of the SGS and EGS dates should be thoroughly evaluated around validation sites by analysing satellite observations with a fine spatial resolution (e.g., 10 m) [31]. In particular, to evaluate the long-term effects of climate change, we need to examine the spatiotemporal variability of SGS and EGS dates for at least the past 120 years, during which continuous observations of temperature have been recorded with modern-day technology. Making connections between plots (by in situ observations) and across regional scales (by satellite observations) [37] as well as performing century-scale phenological observations at the community scale [38] are especially important tasks that have been carried out to a limited extent in previous studies.

Accordingly, we calculated the SGS and EGS dates from 1900 to 2020 by using a degree-day phenological model that was developed for a cool-temperate, deciduous, broad-leaved forest

site located in the upper reaches of the basin [32]. We then assessed the accuracy of the modelled dates by using daily images of vegetation phenology taken in the upper reaches of the basin [39,40], daily live camera images taken in the lower reaches [41], and satellite observations with a spatial resolution of 10 m. The goals of this study were (1) to examine the characteristics of the spatiotemporal variability of the SGS and EGS dates in a steep river basin under the influence of climate change on a century timescale and to identify what caused the variability, and (2) to assess the generality and representativeness of a degree-day model of vegetation phenology.

## 2. Materials and methods

### 2.1. Study region and sites

Our target was the basin of the Daihachiga River in Takayama, Gifu prefecture. The basin is located in the mountainous region of central Japan (Fig 1). The elevation of the basin ranges from 1595 m a.s.l. at the top to 560 m a.s.l. at the bottom [42]. The basin is about 20 km from east to west and about 5 km from north to south (137˚15′E, 36˚10′N; 137˚27′E, 36˚10′N; 137˚15′E, 36˚6′N; 137˚27′E, 36˚6′N). Its total area is about 60 km$^2$ [37,43,44]. We used a previously developed degree-day phenological model [32] to calculate the SGS and EGS dates in the upper reaches of the basin at a cool-temperate, deciduous, broad-leaved forest site, Takayama (TKY; 36˚08′46″N, 137˚25′23″E, 1420 m a.s.l.; [37,45]), based on data collected at the Takayama Automated Meteorological Data Acquisition System (AMeDAS) weather station (Takayama-AMeDAS; 36˚09′22″N, 137˚15′12″E, 560 m a.s.l.; Fig 1B), about 300 m southwest of the basin. At TKY, there is an eco-tower in which camera systems that monitor vegetation phenology have been installed [37] (TKY-tower; Fig 1B). The annual temperature and precipitation averaged 6.5˚C and 2089 mm, respectively, at TKY during 1996–2009 [45] and 11.0˚C and 1700 mm, respectively, at Takayama-AMeDAS during 1981–2010 (https://www.data.jma.go.jp/obd/stats/etrn/index.php, accessed 6 July 2021). At Takayama-AMeDAS, temperatures have been recorded since 11 May 1899. This site is one of 59 long-term, historical weather stations in Japan where temperature data have been archived since 1901 [46].

The typical landscape within and around the Daihachiga River basin consists of deciduous, broad-leaved forests (mainly deciduous oak and birch), deciduous, coniferous forests (larch), and evergreen, coniferous forests (mainly Japanese cedar and cypress). Deciduous, broad-leaved forests are distributed mainly on the southern slope at elevations of 800–1400 m a.s.l. [48]. At TKY, the dominant canopy tree species are *Quercus crispula* (deciduous oak) and *Betula ermanii* (birch), with some *Fagus crenata* (beech), *Betula platyphylla* var. *japonica* (birch), and *Magnolia obovata* (magnolia). The sub-dominant canopy tree species are *Acer distylum* (lime-leaved maple), *Acer rufinerve* (snakebark maple), *Acanthopanax sciadophylloides*, *Tilia japonica* (Japanese linden), *Sorbus alnifolia* (Korean mountain ash), and *Kalopanax pictus* (castor aralia). The dominant shrub tree species are *Hydrangea paniculata* (panicled hydrangea) and *Viburnum furcatum* (forked viburnum). The forest floor is fully covered by an evergreen dwarf bamboo (*Sasa senanensis*; striped bamboo) [49,50].

### 2.2. Daily mean air temperature

Air temperatures are recorded by AMeDAS stations installed by the Japan Meteorological Agency at about 840 sites in Japan. The interval between sites is about 21 km (https://www.jma.go.jp/jma/kishou/know/amedas/kaisetsu.html, accessed 6 July 2021). However, there are only 12 AMeDAS stations at elevations above 1000 m a.s.l. [13]. It is therefore difficult to obtain temperature data at multiple points in a steep river basin such as the Daihachiga River basin. In fact, there was only one AMeDAS station within our target area (i.e., the Takayama-

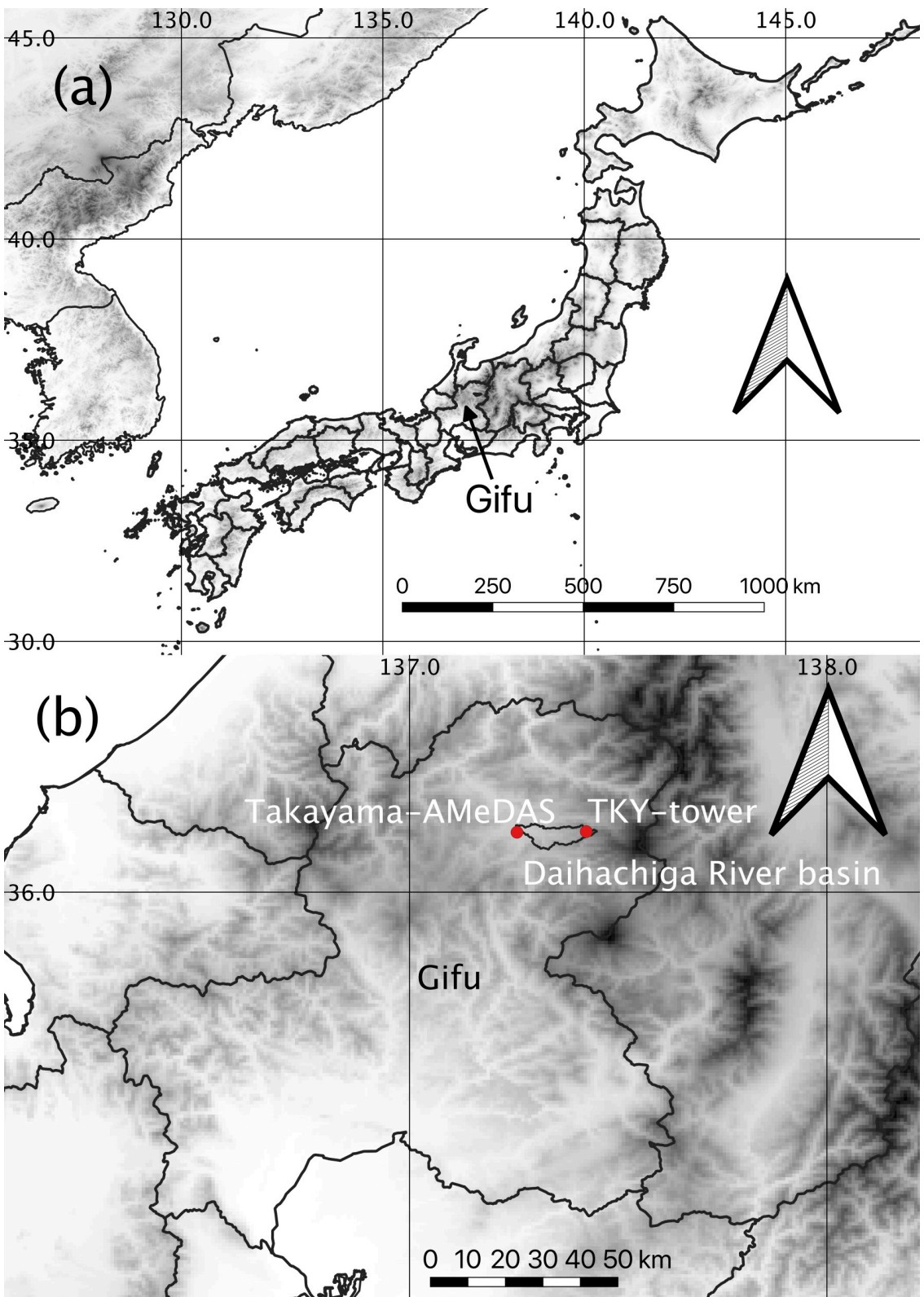

**Fig 1. Summary of target steep river basin and sites.** Horizontal and vertical lines are latitude and longitude lines. (a) Location in Takayama, Gifu prefecture, central Japan. (b) Locations in the Daihachiga River basin, cool-temperate, deciduous broad-leaved forest site: Takayama (TKY; 1420 m a.s.l.), and Takayama AMeDAS weather station (Takayama-AMeDAS; 560 m a.s.l.). Elevation is

indicated in grey scale (white: 0 m; black: 3000 m a.s.l.). We used the GTOPO30 digital elevation model (https://www.usgs.gov/centers/eros/science/usgs-eros-archive-digital-elevation-global-30-arc-second-elevation-gtopo30?qt-science_center_objects=0#qt-science_center_objects, accessed on 6 July 2021). We processed and used the numerical data on the land of a country (administrative district data) published by the Ministry of Land, Infrastructure, Transport and Tourism of Japan ([47], https://nlftp.mlit.go.jp/ksj/gml/datalist/KsjTmplt-N03-v2_4.html, accessed on 6 July 2021). We used the "10m cultural vectors" (https://www.naturalearthdata.com/downloads/10m-cultural-vectors/, accessed on 6 July 2021).

AMeDAS station). However, meteorological observations have been conducted at a weather station about 500 m south of the TKY-tower (TKY-weather station; 36°08′34″N 137°25′20″E, 1342. a.s.l.) since January 1996 ([32]; https://www.green.gifu-u.ac.jp/takayama/, accessed 6 July 2021).

We assumed that daily mean temperature data were not typically available at high elevations. We estimated daily mean temperature at TKY-tower by using a temperature lapse rate of 0.6°C $(100 \text{ m})^{-1}$ [15,26,51,52] and daily mean temperatures observed at Takayama-AMeDAS from 1 January 1990 to 31 May 2020 (https://www.data.jma.go.jp/gmd/risk/obsdl/index.php, accessed 6 July 2021). We assumed that the heat island effect on temperature caused by urbanization was relatively low at Takayama-AMeDAS. Despite merging of municipalities in 2005, the population of the city of Takayama increased from 63,520 in 1920 to only 89,182 in 2015 (https://www.city.takayama.lg.jp/_res/projects/default_project/_page_/001/011/771/18gou_siryou2.pdf, accessed 6 July 2021). In the Discussion, we consider the effect on the calculated SGS and EGS dates of the seasonal gap between the estimated daily mean temperature at TKY-tower and the daily mean temperature that was actually observed at TKY-weather station.

To evaluate the characteristics of climate change, we examined the timeseries of 30-year monthly mean temperatures (climatological mean) and their standard deviations (SDs) during March–May and September–November at Takayama-AMeDAS calculated at intervals of 10 years. In accordance with the Japanese Meteorological Agency, the climatological mean was defined as the average during 30 years at intervals of 10 years. We assumed that the year-to-year variability of monthly mean temperatures (i.e., the SDs) had an important effect on the modelled SGS and EGS dates.

## 2.3. Degree-day model of vegetation phenology

We used a degree-day model of vegetation phenology that we developed by analysing the relationship between daily mean temperature and daily phenological images from 2004 to 2011 at TKY-tower [32]. The timing of leaf flush (i.e., SGS date) as well as leaf colouring and leaf fall (i.e., EGS date) in Japan are explained mainly by temperatures [8,9,53,54]. The SGS date was defined as the first day when the cumulative effect temperature (CET) was greater than 255.4°C. This $CET_{\text{SGS}}$ was calculated with Eq (1), where we set the start date to be 1 January and the threshold temperature for the $CET$ ($T_{\text{t,SGS}}$) to be 2°C. In contrast, the EGS date was defined as the first day when the CET was less than −375.1°C. This $CET_{\text{SGS}}$ was calculated with Eq (2), where we set the start date to be 1 August and the threshold temperature for $CET$ ($T_{\text{t,EGS}}$) to be 18°C [32].

$$CET_{\text{SGS}} = \sum_{i=1}^{D_{\text{SGS}}} \max(T_i - T_{\text{t,SGS}}, 0) \tag{1}$$

$$CET_{\text{EGS}} = \sum_{i=213 \text{ or } 214}^{D_{\text{EGS}}} \min(T_i - T_{\text{t,EGS}}, 0) \tag{2}$$

We assumed that there was no phenological plasticity associated with climate change [5,25,55]. The leaf flush process was explained by a chilling requirement for the release of bud

dormancy and a heating requirement after the release of bud dormancy [34,56,57]. However, we did not incorporate a chilling requirement for the release of bud dormancy, because the basin of the Daihachiga River is sufficiently cold in winter (December to March) to satisfy the requirement for a dormancy period. Climatological means during winter ranged from −1.4˚C in January to +2.9˚C in March (https://www.data.jma.go.jp/obd/stats/etrn/index.php, accessed 6 July 2021). We therefore expected to reduce the degrees of freedom of the degree-day model of vegetation phenology by not including a chilling requirement.

First, we calculated the SGS dates from 1900 to 2020 at TKY-tower and the EGS dates from 1900 to 2019 at Takayama-AMeDAS. Second, we examined the long-term linear trends from 1900 to 2019/2020 and timeseries of 30-year monthly SDs, calculated at intervals of 10 years, of modelled SGS and EGS dates at TKY-tower, Takayama-AMeDAS, and their difference. The climatological mean of temperatures was defined as the 30-year average temperature at intervals of 10 years. Finally, to evaluate the effect of climate change, we examined the correlations between the SDs of 30-year monthly mean temperatures at timesteps of 10 years and the SDs of 30-year modelled SGS and EGS dates at TKY-tower, Takayama-AMeDAS, and their difference.

## 2.4. Camera images

To assess the generality of the modelled SGS and EGS dates, we used daily images of phenology taken at TKY-tower [39,40] and live camera images taken at two points near Takayama-AMeDAS: Kaji-bridge, 1.5 km south (36˚8′367″N, 137˚15′27″E, 572 m a.s.l.; Figs 7 and 8; [42]), and Koito-Pottery, 2.6 km north-east (36˚8′8″N, 137˚14′24″E, 605 m a.s.l.; Figs 7 and 8, [42]) of Takayama-AMeDAS (Nagai et al., 2018b). At TKY-tower, daily images of phenology have been publicly available since 2003 on the web site of the Phenological Eyes Network (http://www.pheno-eye.org, accessed 6 July 2021). The daily images of phenology showed mainly *Q. crispula*, *B. ermanii* (dominant canopy tree species), *A. distylum* (sub-dominant canopy tree species), and *Prunus maximowiczii* [58,59]. At Koito-Pottery and Kaji-bridge, the latest live camera images are publicly available on an hourly or 3-minute timeframe from the Web sites of the Koito Pottery (https://koitoyaki.com/livecam/, which we have not accessed since March 2021) and Takayama Printing Co, Ltd (https://www.takayama-dp.com/live/, accessed 6 July 2021). At Koito-Pottery, the live camera images showed mainly deciduous trees close to the cameras and distant deciduous and evergreen forests. At Kaji-bridge, the live camera images showed mainly deciduous trees close to the cameras and distant deciduous forests. We assumed that there was no difference between the SGS and EGS dates at Takayama-AMeDAS, Koito-Pottery, and Kaji-bridge, because the difference in elevation among the three sites is very small (max. 45 m). We have downloaded live camera images by running shell scripts and the *crontab* command on a Linux personal computer since December 2017. We obtained permission to download images from the Koito Pottery and Takayama Printing Co, Ltd.

## 2.5. Satellite data

To assess the spatial representativeness of the modelled SGS and EGS dates, we used surface reflectance data in the red, green, and blue (RGB) bands with a 10-m spatial resolution observed by SENTINEL-2A/2B satellites (https://sentinel.esa.int/web/sentinel/missions/sentinel-2, accessed 6 July 2021). The modelled SGS and EGS dates may have been affected by the spatial heterogeneity and distribution of tree species as well as by microtopography. For this reason, analysis of SENTINEL-2A/2B satellite observations was a useful way to fill spatial gaps between modelled dates in a degree-day model of phenology and in situ observed data at a validation site [33]. We collected data observed within 5 days before and after modelled SGS

and EGS dates at TKY-tower and Takayama-AMeDAS because the time interval between SENTINEL-2A and 2B satellite observations is 5 days (by both satellites). Despite the short observation period of the SENTINEL-2A/2B satellites (since 2015 for SENTINEL-2A and 2017 for SENTINEL-2B), the SENTINEL-2A and 2B satellites are the only non-commercial satellites that have provided accurate geographical distributions of plant phenology in a steep river basin with a high spatiotemporal resolution.

We downloaded satellite data from the CREODIAS web site (https://creodias.eu, accessed 6 July 2021). We collected data since 2018 because the SENTINEL-2A and 2B satellites were launched in 2015 and 2017, respectively. We collected only data with less than or equal to 20% cloud coverage. We used RGB composite images and calculated the green-red vegetation index (GRVI; [60]). In deciduous broad-leaved forests, a GRVI of 0 indicates both the start of leaf flushing [60] and the time of maximum leaf colouring and leaf fall [61]. In this study, atmospheric corrections were performed by using the SNAP (Sentinel Application Platform) toolboxes (http://step.esa.int/main/toolboxes/snap/, accessed 6 July 2021). We did not apply further cloud masking to the satellite data because there was no cloud contamination in our target river basin.

We conducted all analyses with Apache OpenOffice 4.1.7 (https://www.openoffice.org, accessed 6 July 2021), LibreOffice 7.0.3.1 (https://ja.libreoffice.org, accessed 6 July 2021), R v. 3.6.2 [62], and QGIS v. 3.1 (https://qgis.org/ja/site/, accessed 6 July 2021) software and shell scripts.

## 3. Results

### 3.1. Modelled SGS and EGS dates

The modelled SGS dates ranged from DOY 108 to 132 (average DOY 120) at Takayama-AMeDAS and from DOY 124 to 161 (average DOY 146) at TKY-tower (Fig 2A). Despite interannual fluctuations, the dates advanced by an average of 0.98 day decade$^{-1}$ ($R^2 = 0.33$, $p < 0.001$) at TKY-tower and by 0.66 day decade$^{-1}$ ($R^2 = 0.17$, $p < 0.001$; Fig 2A) at Takayama-AMeDAS. The difference in SGS dates between sites ranged from 16 to 36 days (average 26 days; Fig 2C). Despite interannual fluctuations, this difference decreased by an average of 0.32 day decade$^{-1}$ ($R^2 = 0.07$, $p < 0.01$; Fig 2C). The modelled EGS dates ranged from DOY 284 to 309 (average DOY 297) at TKY-tower and from DOY 312 to 333 (average DOY 321) at Takayama-AMeDAS (Fig 2B). Despite interannual fluctuations, the dates were delayed by an average of 0.75 day decade$^{-1}$ ($R^2 = 0.26$, $p < 0.001$) at TKY-tower and by 0.72 day decade$^{-1}$ ($R^2 = 0.28$, $p < 0.001$; Fig 2B) at Takayama-AMeDAS. The difference in dates between sites ranged from 18 to 33 days (average 25 day; Fig 2D). There was no significant long-term trend in the difference of EGS dates between sites (Fig 2D).

The sensitivity of the modelled SGS and EGS dates to elevation changed from 3.29 days (100 m)$^{-1}$ (−5.48 day ˚C$^{-1}$) and −2.89 days (100 m)$^{-1}$ (4.81 day ˚C$^{-1}$), respectively, in 1900 to 2.85 days (100 m)$^{-1}$ (−4.75 day ˚C$^{-1}$) and −2.84 days (100 m)$^{-1}$ (4.73 day ˚C$^{-1}$), respectively, in 2019 (Fig 2).

The SDs of the modelled SGS dates during 1940−1969 and 1980−2009 at TKY-tower and Takayama-AMeDAS, respectively, were large compared to the corresponding SDs during 1900−1929 and 1910−1939, respectively. The SD of the differences in dates between sites was larger from 1950−1979 to 1990−2019 than from 1900−1929 to 1920−1959 (Fig 3A). In contrast, the SDs of the modelled EGS dates at TKY-tower and Takayama-AMeDAS were larger from 1960−1989 to 1990−2019 than from 1900−1929 to 1940−1969. The interannual variability of the SDs of the difference in EGS dates between sites was smaller than that of the SDs of the modelled EGS dates at the sites (Fig 3B).

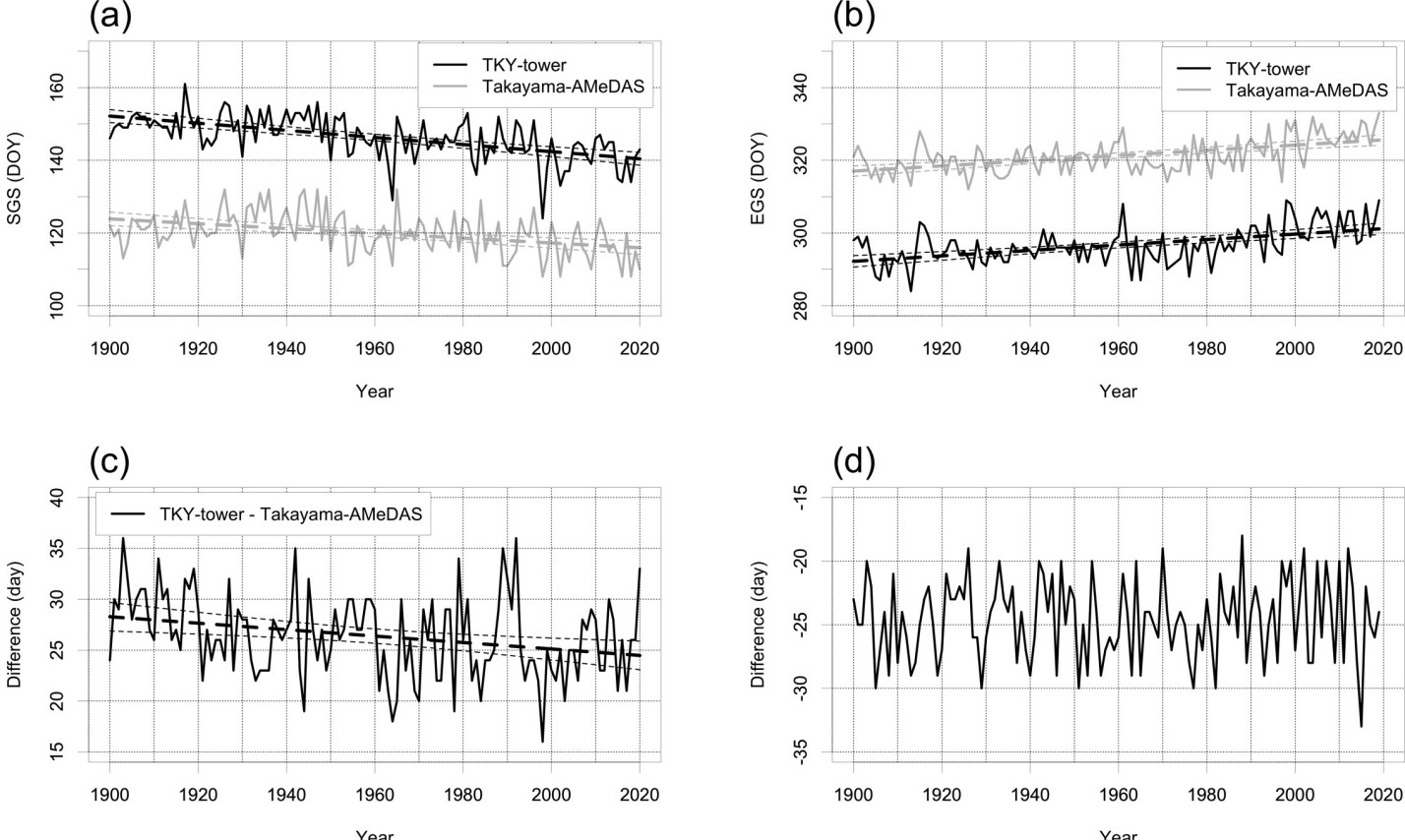

**Fig 2.** Time-series of modelled (a) SGS and (b) EGS dates at TKY-tower (1420 m a.s.l.) and Takayama-AMeDAS (560 m a.s.l.). (c, d) Differences between sites in modelled (c) SGS and (d) EGS dates (Takayama-AMeDAS dates minus TKY-tower dates). The dashed lines show statistically significant ($p < 0.05$) linear trends and their 95% confidence intervals.

### 3.2. Air temperature

Timeseries of 30-year monthly mean temperatures, calculated at intervals of 10 years (Fig 4A and 4C), showed significantly positive rates of change in March, April, May, September, October, and November (Table 1).

Compared with May, the SD of the monthly mean temperature in March and April was always large (Fig 4B). In contrast, compared with September and October, the SD of the 30-year monthly mean temperature in November was large from 1900–1929 to 1930–1959. In addition, the SDs have gradually increased since 1930 (Fig 4D).

### 3.3. Modelled SGS and EGS dates and temperature relationships

Table 2 shows the relationship between the SDs of the 30-year monthly mean temperatures, calculated at intervals of 10 years (Fig 4B and 4D), and the 30-year modelled SGS (Fig 3A) and EGS (Fig 3B) dates at TKY-tower, Takayama-AMeDAS, and their difference. At TKY-tower, the SDs of the 30-year modelled SGS dates were correlated significantly with those of the monthly mean temperatures in April and May. In contrast, at TKY-tower and Takayama-AMeDAS, the SDs of the 30-year modelled EGS dates were significantly correlated with those of the monthly mean temperatures in September and October.

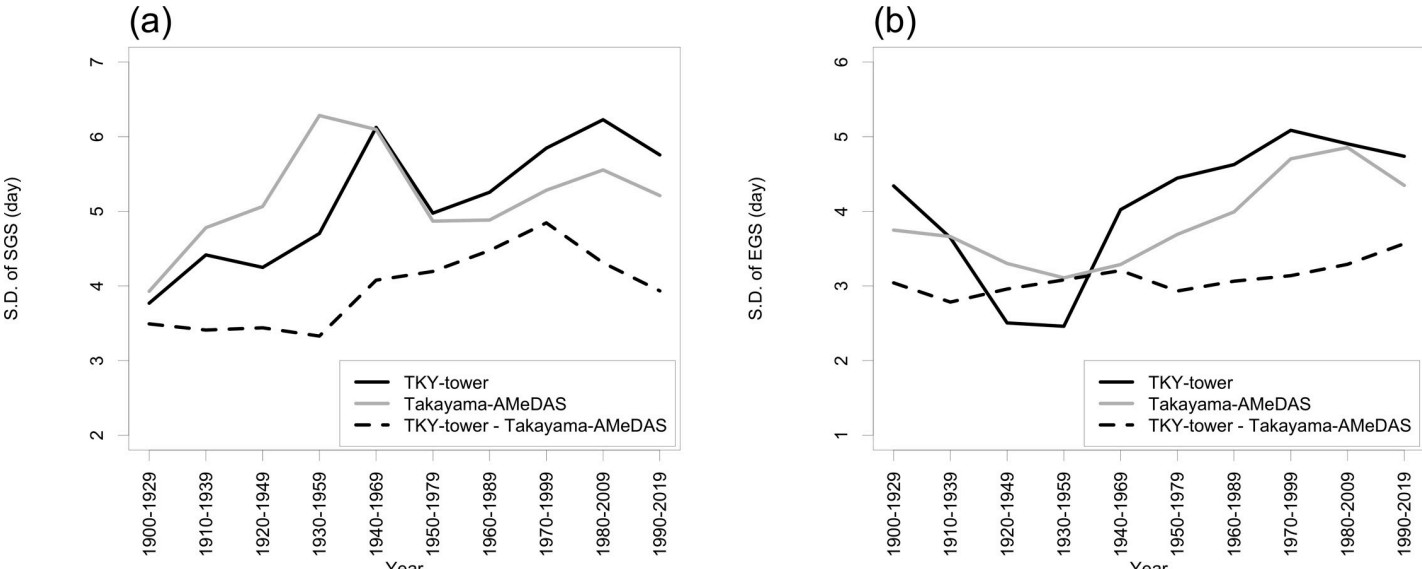

**Fig 3.** Time-series of 30-year SDs of modelled (a) SGS and (b) EGS dates at TKY-tower, Takayama-AMeDAS, and their difference (Takayama-AMeDAS dates minus TKY-tower dates).

### 3.4. Assessment via camera images and satellite observations

At TKY-tower, images of phenology taken on modelled SGS dates in each year showed almost the same condition of the canopy surface (Fig 5A). This condition was almost the same as the condition that defined the SGS date based on an analysis of RGB values extracted from images of phenology by Nagai et al. (2013) [32, Fig 2]. In addition, the modelled SGS dates from 2004 to 2011 correlated significantly with the observed SGS dates shown in Table 1 of Nagai et al. (2013) (Spearman's rank correlation coefficient, $\rho$ = 0,94, $p$ < 0.001) [32]. In contrast, images taken on modelled EGS dates in each year showed small variations in the conditions of leaf colouring and leaf fall on the canopy surface (Fig 5B). However, those images included the same conditions as the images of phenology associated with the EGS date defined by Nagai et al. (2013) [32, Fig 2]. In addition, the modelled EGS dates from 2004 to 2011 correlated significantly with the observed EGS dates shown in Table 1 of Nagai et al. (2013) (Spearman's rank correlation coefficient, $\rho$ = 0.97, $p$ < 0.001) [32]. At Koito-Pottery and Kaji-bridge, despite variations among individual trees, live camera images taken on the modelled SGS dates at Takayama-AMeDAS in 2018 and 2019 showed conditions just around the time of leaf flush. Those images in 2020 showed conditions about 10 days before the time of leaf flush (Fig 6A). In contrast, despite variations among individual trees, images taken on the modelled EGS dates at Takayama-AMeDAS showed leaf colouring and leaf fall (Fig 6B).

Table 3 shows the relationships between the modelled SGS and EGS dates and the closest satellite-observed dates. Except for 2018, we could not obtain satellite data with less than or equal to 20% cloud coverage at about the times of the four modelled dates. The satellite-observed dates were 2 or 3 days later than the modelled SGS and EGS dates. On the satellite-observed date closest to the modelled SGS date at Takayama-AMeDAS (3 days later than the modelled SGS date), leaf flush had not occurred widely throughout the river basin (Fig 7A). The GRVI of deciduous, broad-leaved forests was less than or equal to 0.0 (Fig 7B). On the satellite-observed date closest to the modelled SGS at TKY-tower (2 days later than the modelled SGS date), leaf flush had expanded throughout the basin (Fig 7C), and the GRVI in deciduous, broad-leaved forests had increased to 0.2–0.4. In addition, the GRVI around TKY-tower had

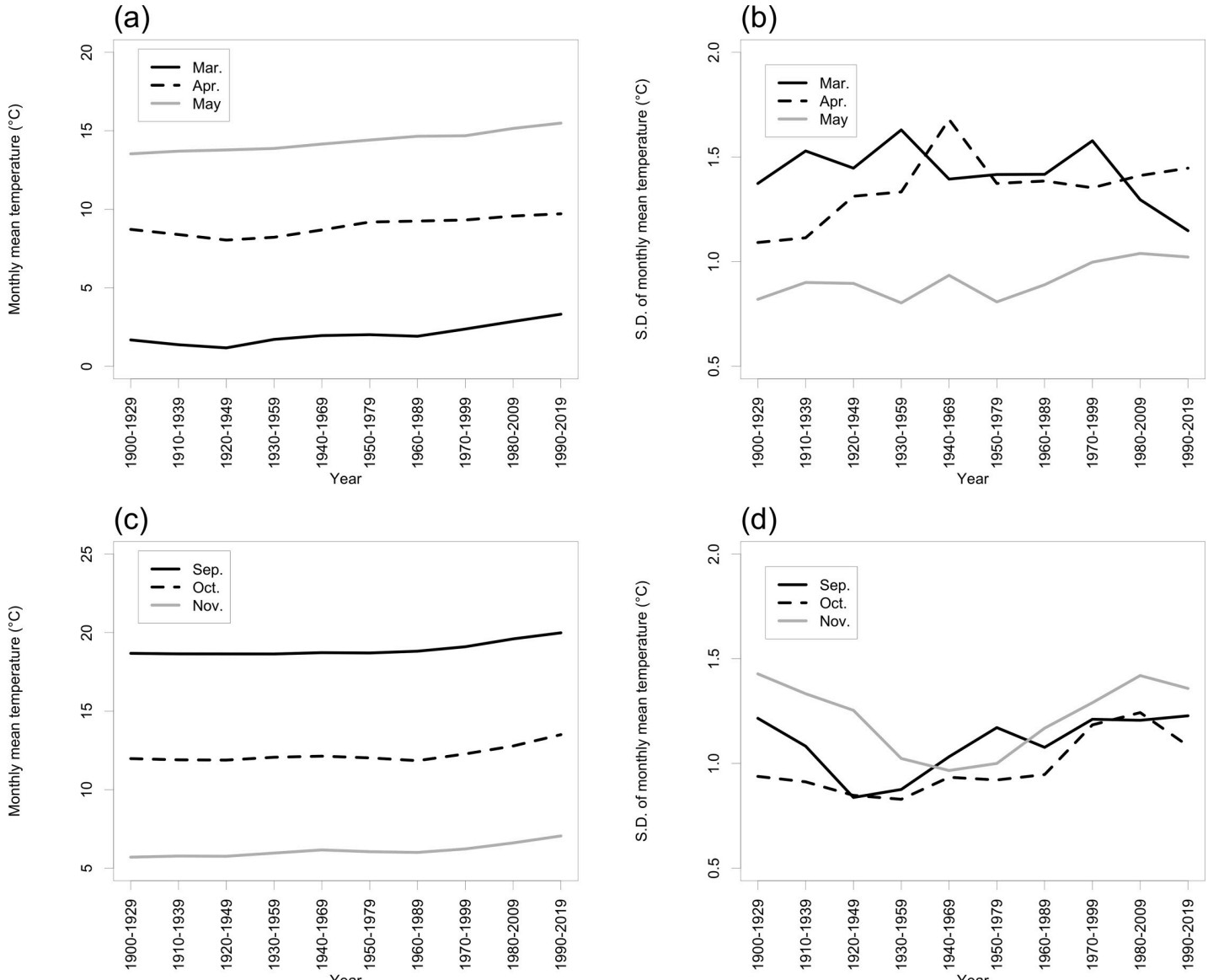

**Fig 4.** Time-series of (a, c) 30-year monthly mean temperatures and (b, d) SDs during (a, b) March–May and (c, d) September–November at Takayama-AMeDAS. Table 1 shows the rates of change of the 30-year monthly mean temperatures and their statistical significance.

**Table 1. Summary of the rates of change of the 30-year monthly mean temperatures at a timestep of 10 years (Fig 4A and 4C) at Takayama-AMeDAS and their significance based on Pearson's correlation coefficient.**

| Month | Trend (°C year$^{-1}$) | *p*-value |
|---|---|---|
| Mar. | 0.019 | $p < 0.001$ |
| Apr. | 0.017 | $p < 0.01$ |
| May | 0.022 | $p < 0.001$ |
| Sep. | 0.013 | $p < 0.01$ |
| Oct. | 0.013 | $p < 0.05$ |
| Nov. | 0.012 | $p < 0.001$ |

**Table 2. Summary of correlations (Spearman's rank, ρ) between SDs of 30-year monthly mean temperature at a time-step of 10 years (Fig 4B and 4D) and that of 30-year modelled SGS (Fig 3A) and EGS dates (Fig 3B) at TKY-tower, Takayama-AMeDAS, and their difference.**

| Phenology | Location | Month | ρ | *p*-value |
|---|---|---|---|---|
| SGS | TKY-tower | Mar. | −0.31 | 0.39 |
| | | Apr. | 0.87 | <0.01 |
| | | May | 0.68 | <0.05 |
| | Takayama-AMeDAS | Mar. | 0.14 | 0.71 |
| | | Apr. | 0.56 | 0.10 |
| | | May | 0.27 | 0.45 |
| | TKY-tower − Takayama-AMeDAS | Mar. | −0.24 | 0.51 |
| | | Apr. | 0.52 | 0.13 |
| | | May | 0.42 | 0.23 |
| EGS | TKY-tower | Sep. | 0.73 | <0.05 |
| | | Oct. | 0.95 | <0.001 |
| | | Nov. | 0.37 | 0.30 |
| | Takayama AMeDAS | Sep. | 0.76 | <0.05 |
| | | Oct. | 0.93 | <0.001 |
| | | Nov. | 0.62 | 0.06 |
| | TKY-tower − Takayama-AMeDAS | Sep. | 0.35 | 0.33 |
| | | Oct. | 0.64 | 0.05 |
| | | Nov. | 0.15 | 0.68 |

increased to 0.1–0.4 (Fig 7D). In contrast, on the satellite-observed date closest to the modelled EGS at TKY-tower (3 days later than the modelled EGS date), leaf colouring and leaf fall had occurred at elevations greater than 1000 m (Fig 8A). The GRVI in deciduous, broad-leaved forests was less than or equal to 0.0 (Fig 8B). On the satellite-observed date closest to the modelled EGS at Takayama-AMeDAS (2 days later than the modelled EGS date), leaf colouring and leaf fall had advanced throughout the basin (Fig 8C), and the area characterized by GRVIs less than or equal to 0.0 had expanded in the basin (Fig 8D).

## 4. Discussion

### 4.1. Sensitivity of modelled SGS and EGS dates to elevation

The results of our analysis of the sensitivity of modelled SGS and EGS dates to elevation (Fig 2) implied the following relationships. The modelled SGS date at Takayama-AMeDAS (560 m a. s.l.) in 1900 (DOY 124) was equivalent to the SGS date at 843 m a.s.l. in 2020 (shift to 283 m higher elevation). The modelled EGS date at Takayama-AMeDAS (560 m a.s.l.) in 1900 (DOY 317) was equivalent to the EGS at 861 m a.s.l. in 2019 (shift to 301 m higher elevation). However, the yearly difference of the modelled SGS dates at Takayama-AMeDAS and TKY ranged from −15 days (earlier than previous year) to 21 days (later than previous year) and from −16 days to 23 days, respectively (Fig 2A). These differences are equivalent to elevation changes of −526 m (shift to lower elevation) to +737 m (shift to higher elevation) and to elevation changes of −561 m to +807 m, respectively (in the case of 2019). The yearly difference of the modelled EGS dates at Takayama-AMeDAS and TKY ranged from −10 days (earlier than previous year) to 17 days (later than previous year) and from −12 days to 15 days, respectively (Fig 2B). These differences were equivalent to elevation changes of +352 m to −599 m and +423 m to −528 m, respectively (in the case of 2019). These comparisons suggest that deciduous, broad-leaved forests in the Daihachiga River basin might have acclimated to the year-to-year variability of climate during the past 120 years by changing their leaf flush and leaf fall phenology. In other

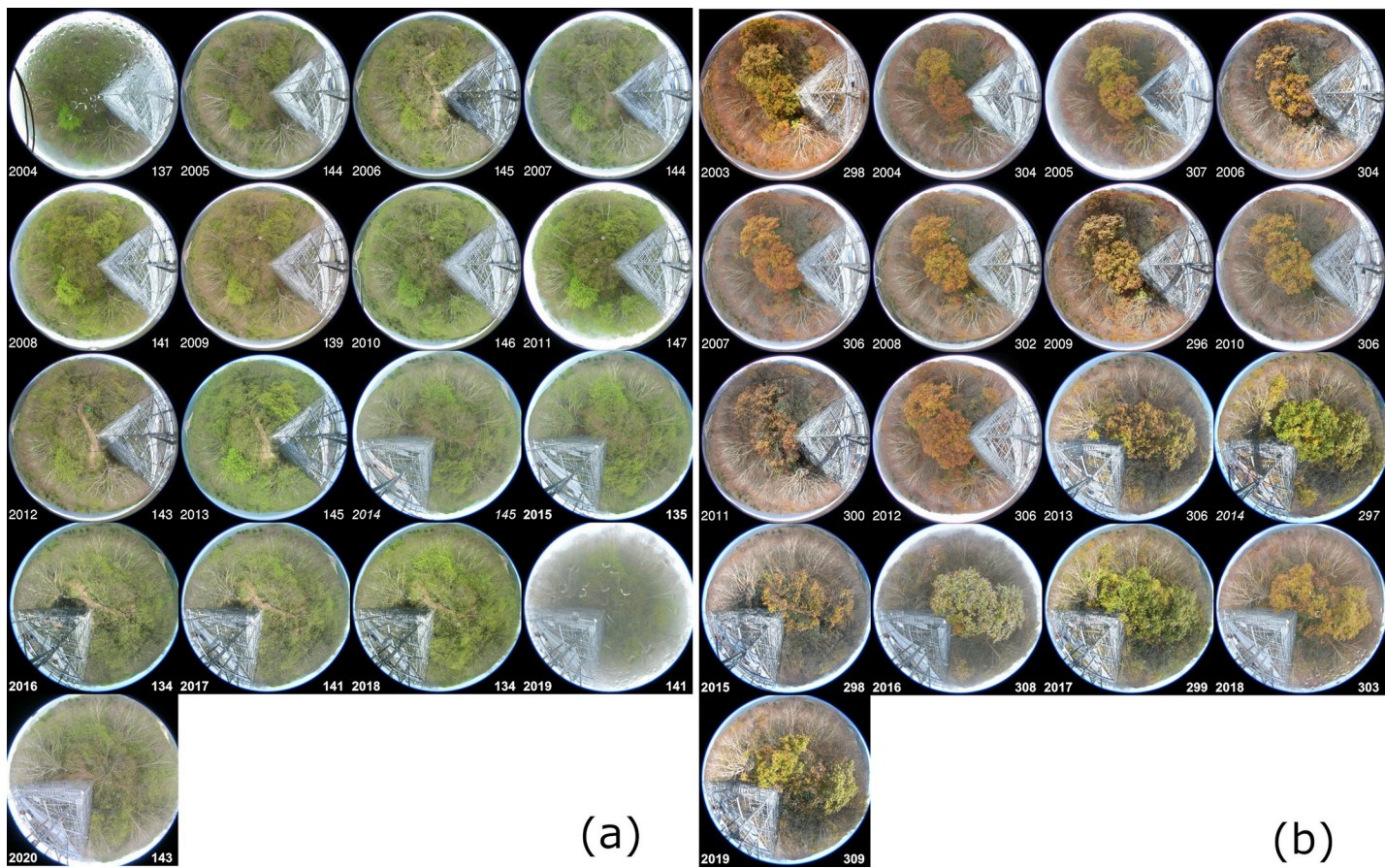

**Fig 5.** Images of phenology taken on modelled (a) SGS and (b) EGS dates at TKY-tower, in the upper reaches of the Daihachiga River basin. The year and DOY (day of year) are shown in the bottom-left and bottom-right corners, respectively, of each image.

words, the warmer spring temperature triggered earlier SGS dates, and the warmer autumn temperature triggered later EGS dates. However, this acclimation may not apply to other river basins where annual mean temperatures are much higher than the temperatures of the Daihachiga River basin. Nagai et al. (2020) [63] reported that the correlation between the observed first flowering date of Yoshino cherry and latitude has decreased since 1980 in Japan. They explained the lower correlation by reasoning that higher temperatures in winter had delayed the release from dormancy and the subsequent first-flowering dates in areas where the annual mean temperature was high. To evaluate the significance of the chilling requirement for dormancy release, we will need to examine the relationships between elevation and the dates of SGS and EGS in other steep river basins where annual mean temperatures are much higher than the temperatures in the Daihachiga River basin.

Our result was similar to results previously reported for the SGS dates of oak and larch and for the flowering of horse chestnut, alder, cocksfoot grass, goat willow, rye, and small-leaved lime (Table 4). The sensitivity we observed, however, was large compared with the sensitivity of the flowering of Norway spruce and common wine grapes, the SGS of beech, and other SGS and EGS dates inferred from satellite observations (absolute values) (Table 4). The following reasoning may account for these similarities and differences. The degree-day phenology model that we used [32] was developed at TKY, where the dominant tree species are oak (*Q. crispula*) and birch (*B. ermanii* and *B. platyphylla*) [37,49,50]. Consequently, our results were similar to

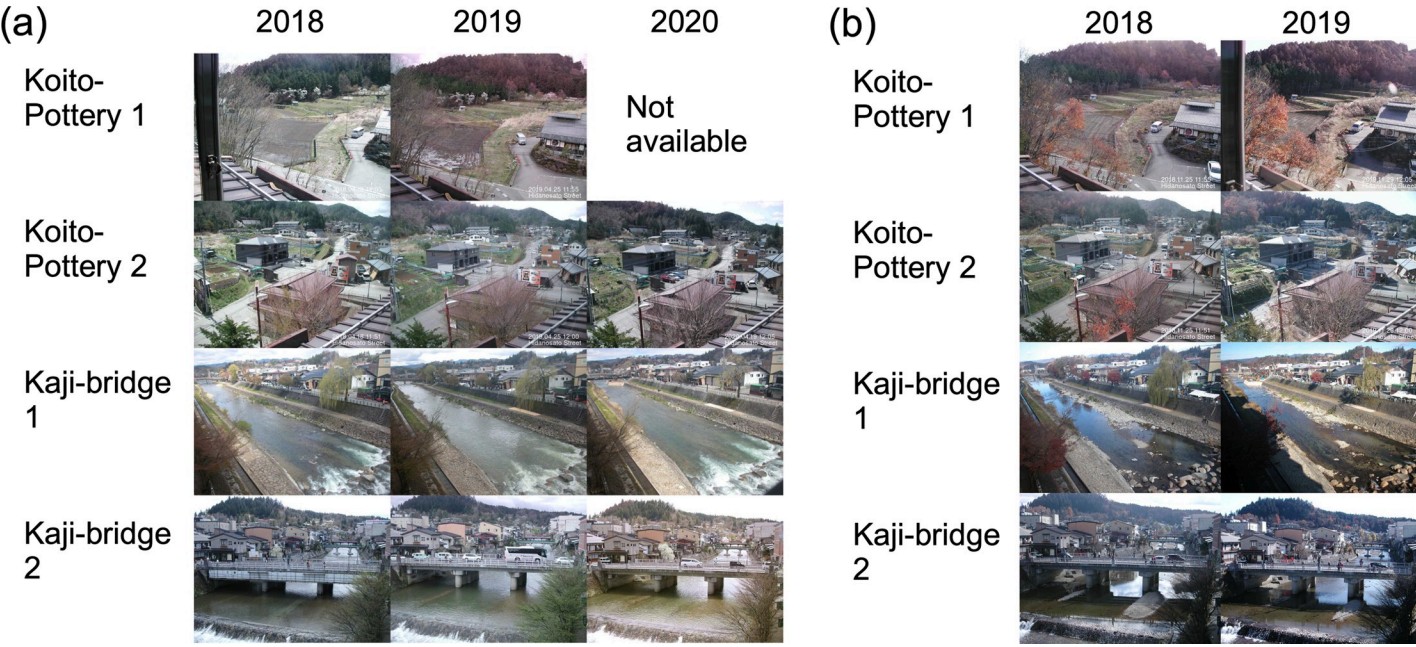

**Fig 6.** Images of phenology taken on modelled (a) SGS and (b) EGS dates at Koito-Pottery and Kaji-bridge in the lower reaches of the Daihachiga River basin.

**Fig 7.** (a, c) RGB composite images and (b, d) GRVI observed within 5 days before and after modelled SGS dates at (a, b) TKY-tower and (c, d) Takayama-AMeDAS in 2018. Solid line shows the boundary of the Daihachiga River basin. Strange colour is shown in the top of image in Koito-Pottery 1 in 2019. We used the ALOS Global Digital Surface Model "ALOS World 3D – 30m" (AW3D30) published by Japan Aerospace Exploration Agency (https://www.eorc.jaxa.jp/ALOS/en/aw3d30/index.htm, accessed on 6 July 2021).

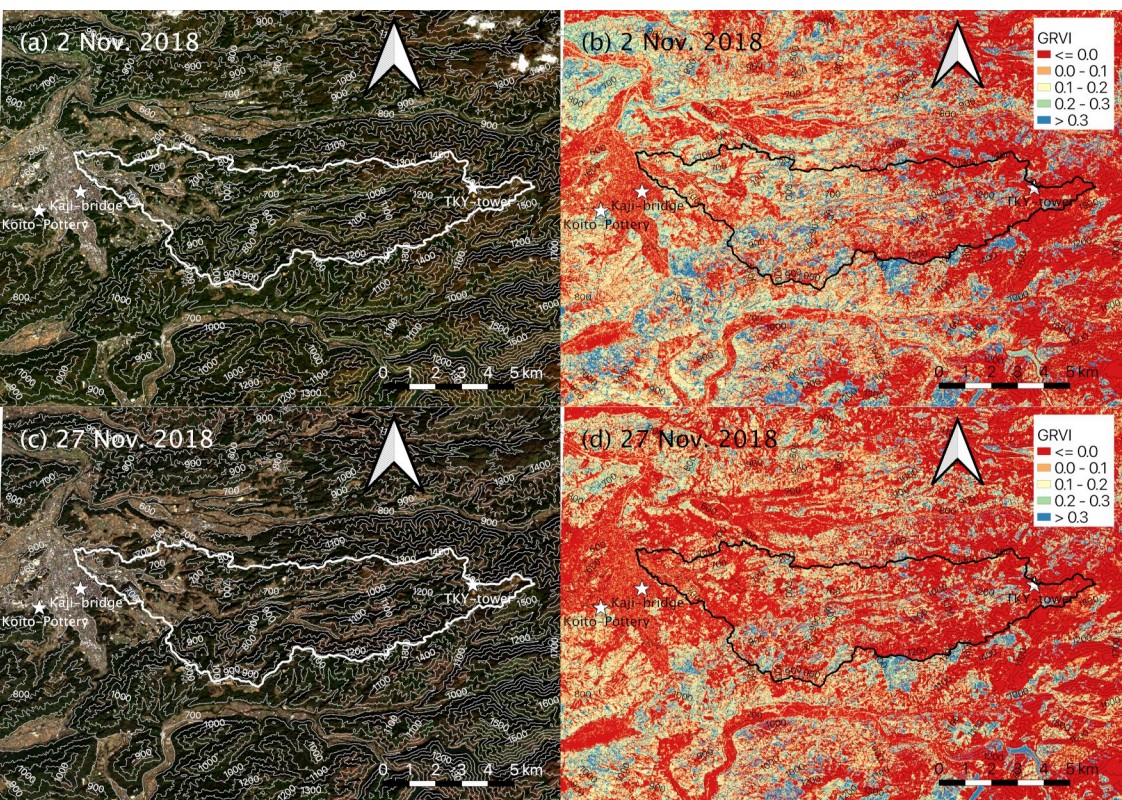

**Fig 8.** (a, c) RGB composite images and (b, d) GRVI observed within 5 days before and after modelled EGS dates at (a, b) TKY-tower and (c, d) Takayama-AMeDAS in 2018. Solid line shows the boundary of the Daihachiga River basin. Strange colour is shown in the top of images in Koito-Pottery 1. We used the ALOS Global Digital Surface Model "ALOS World 3D – 30m" (AW3D30) published by Japan Aerospace Exploration Agency (https://www.eorc.jaxa.jp/ALOS/en/aw3d30/index.htm, accessed on 6 July 2021).

previously reported results for oak. In contrast, the timing and patterns of leaf flush and leaf fall in beech, which is sparsely distributed around TKY-tower, differs from those of oak and birch. For instance, the leaf flush of beech is earlier than that of oak and birch [64]. These phenological characteristics may explain the different sensitivities of the SGS and EGS dates to elevation. The characteristics of the sensitivities of flowering, leaf flush, and leaf fall to temperature have been reported in many previous studies [10,11,65,66]. The differences in reported sensitivities suggest that use of a single degree-day model of phenology to estimate SGS and EGS dates may be misleading with respect to those dates in a steep river basin where there is high diversity and an uneven distribution of tree species.

The following lines of reasoning may account for the differences between our results and previously reported satellite-based results of the sensitivity of SGS and EGS dates to elevation. First, satellite sensors observe average plant phenology within a square with sides of 500–1000

**Table 3. Summary of relationships between the modelled SGS and EGS dates and satellite data.**

| Phenology | Location | Modelled | Satellite | Difference (modelled − satellite) (days) |
|---|---|---|---|---|
| SGS | Takayama-AMeDAS | 18 Apr. 2018 (DOY 108) | 21 Apr. 2018 (DOY 111) | −3 |
| | TKY-tower | 14 May 2018 (DOY 134) | 16 May 2018 (DOY 136) | −2 |
| EGS | Takayama-AMeDAS | 25 Nov. 2018 (DOY 329) | 27 Nov. 2018 (DOY 331) | −2 |
| | TKY-tower | 30 Oct. 2018 (DOY 303) | 2 Nov. 2018 (DOY 306) | −3 |

**Table 4. Summary of previously reported relationships between plant phenology and elevation.**

| Phenology | Location | Species/plant functional type | Method | Period | Sensitivity | Reference |
|---|---|---|---|---|---|---|
| Bloom | Nagano, Japan | *Erigeron annuus* | In situ observation | 1993−1998 | 3.7−6.0 day (100 m)$^{-1}$ | [27] |
| Full flowering | Switzerland | Horse chestnut | In situ observation | 1971−2000 | 3.35 ± 0.18 day (100 m)$^{-1}$ | [14] in Table 2 |
| Beginning of flowering | Germany | Common alder | In situ observation | 1971−2000 | 2.92 ± 0.28 day (100 m)$^{-1}$ | [14] in Table 2 |
| Beginning of flowering | Austria, Germany | Common hazel | In situ observation | 1971−2000 | 4.56 ± 0.21 day (100 m)$^{-1}$ | [14] in Table 2 |
| Beginning of flowering | Austria | Cocksfoot grass | In situ observation | 1971−2000 | 2.80 ± 0.41 day (100 m)$^{-1}$ | [14] in Table 2 |
| Beginning of flowering | Austria, Germany | Norway spruce | In situ observation | 1971−2000 | 0.92 ± 0.21 day (100 m)$^{-1}$ | [14] in Table 2 |
| First flowers open | Germany | Goat willow | In situ observation | 1971−2000 | 2.51 ± 0.16 day (100 m)$^{-1}$ | [14] in Table 2 |
| Full flowering | Switzerland | Black elder | In situ observation | 1971−2000 | 3.12 ± 0.21 day (100 m)$^{-1}$ | [14] in Table 2 |
| First flowers open | Austria | Rye | In situ observation | 1971−2000 | 3.22 ± 0.39 day (100 m)$^{-1}$ | [14] in Table 2 |
| First flowers open | Germany | Small-leaved lime | In situ observation | 1971−2000 | 4.03 ± 0.24 day (100 m)$^{-1}$ | [14] in Table 2 |
| Full flowering | Switzerland | Common grape wine | In situ observation | 1971−2000 | 1.13 ± 0.26 day (100 m)$^{-1}$ | [14] in Table 2 |
| SGS | Pyrenees (France) | Oak | In situ observation | 2005−2006 | 7.48 day °C$^{-1}$ | [11] |
| SGS | Pyrenees (France) | Beech | In situ observation | 2005−2006 | 2.0 day °C$^{-1}$ | [11] |
| SGS | Pyrenees (France) | Oak | In situ observation | 2005−2007 | 3.2 day (100 m)$^{-1}$ | [20] |
| SGS | Pyrenees (France) | Beech | In situ observation | 2005−2007 | 1.0 day (100 m)$^{-1}$ | [20] |
| SGS | Pyrenees (France) | Area average | In situ observation | 2005−2007 | 2.3 day (100 m)$^{-1}$ | [20] |
| SGS | Großer Falkenstein (Germany) | Beech | Phenology images | 2006−2007 | 2.5 day (100 m)$^{-1}$ | [18] |
| SGS | Nagano, Japan | Larch | In situ observation | 1980−1991 | 2.9 day (100 m)$^{-1}$ | [26] |
| SGS | Slovakia | Beech | Satellite (MODIS) & in situ observation | 2000−2012 | 2.0 day (100 m)$^{-1}$ | [23] |
| SGS | Japan | DBF | Satellite (MODIS) | 2003−2012 | 1.46−2.59 day (100 m)$^{-1}$ | [8] |
| SGS | Japan | DNF | Satellite (MODIS) | 2003−2012 | 0.32−1.48 day (100 m)$^{-1}$ | [8] |
| SGS | Qinghai-Xizang Plateau, China | Grassland | Satellite (MODIS) | 1982−2006 | 7.8 day (100 m)$^{-1}$ | [21] |
| EGS | Japan | DBF | Satellite (MODIS) | 2003−2012 | −1.28 to −0.70 day (100 m)$^{-1}$ | [8] |
| EGS | Japan | DNF | Satellite (MODIS) | 2003−2012 | −1.23 to −0.10 day (100 m)$^{-1}$ | [8] |
| EGS | North-eastern China | DBF | Satellite (MODIS) & in situ observation | 2000−2012 | −1.5 day (100 m)$^{-1}$ | [22] |

DBF, deciduous broad-leaved forest; DNF, deciduous needle-leaved forest.

m. Such a square may include many tree species characterized by a wide range of the timing and patterns of leaf flush and leaf fall. Second, the SGS and EGS dates defined by satellite observations may differ from those defined by in situ observations. Third, data are missing from a time-series of satellite observations when there is cloud contamination and atmospheric noise. Finally, the characteristics of microtopography cannot be detected by satellite sensors with a spatial resolution of 500–1000 m. Miura et al. [67] showed the utility of analysing a time-series of an index of vegetation based on observations by the AHI (Advanced Himawari Imager) sensor mounted on the geostationary Himawari-8 satellite for phenological observations at TKY. Despite its coarse spatial resolution (1000 m), the AHI sensor has a very high temporal resolution (every 2.5 min around Japan). Important tasks for the future include integrative evaluation of calculations of SGS and EGS dates with a degree-day model, analysis of SENTINEL-2A/2B-satellite observations with a fine spatial resolution but low observation frequency, and analysis of Himawari-8 satellite observations with a high observation frequency but coarse spatial resolution.

## 4.2. Interannual variability of modelled SGS and EGS dates

The difference between modelled SGS dates at Takayama-AMeDAS and TKY-tower decreased by 0.32 day decade$^{-1}$ (Fig 2C). Several different mechanisms may have accounted for this pattern. First, the daily mean temperature between March and May increased (Fig 4A). As a result, the date when the daily mean temperature exceeded the threshold temperature for $CET_{SGS}$ ($T_{t,SGS} = 2˚C$) advanced, especially at TKY-tower. Second, the date when the $CET$ reached $CET_{SGS}$ (= 255.4˚C) also advanced. Ziello et al. [14] reported advances of $0.065 \pm 0.028$ day year$^{-1}$ per 100 m for the full flowering of the common alder and $0.049 \pm 0.020$ day year$^{-1}$ per 100 m for the beginning of the flowering of Norway spruce, and delay of $0.025 \pm 0.011$ day year$^{-1}$ per 100 m for the opening of the first flower of black elder in Europe. Our result of advance of 0.0037 day year$^{-1}$ per 100 m for the SGS was smaller than those of common alder and Norway spruce. In contrast, there was no significant long-term trend of the difference between the modelled EGS dates at Takayama-AMeDAS and TKY-tower (Fig 2D). This lack of a trend may have resulted from the fact that the daily mean temperature from September to November increased (Fig 4C). The date when the daily mean temperature was below the threshold temperature for $CET_{EGS}$ ($T_{t,EGS} = 18˚C$) at those sites was therefore delayed. Because of this delay, the date when the $CET$ reached $CET_{EGS}$ (= −375.1˚C) was also delayed.

The difference between the Takayama-AMeDAS and TKY-tower results was larger for the modelled SGS dates than for the modelled EGS dates (Fig 2C and 2D). This disparity may have been caused by the relationship between the interannual variability of the SGS and EGS dates and that of the monthly mean temperature. The interannual variability of the modelled SGS date correlated with that of monthly mean temperature at TKY-tower but not at Takayama-AMeDAS (Table 2). In contrast, at both sites, the interannual variability of the modelled EGS dates correlated with that of monthly mean temperature (Table 2). These relationships suggest that earlier SGS dates in the lower reaches of a river basin may not be synchronized with the earlier SGS dates in the upper reaches of the same basin.

## 4.3. Sensitivity of temperature to elevation

We used a temperature lapse rate of 0.60˚C (100 m)$^{-1}$, which has been commonly used in several previous studies [15,26,51,52]. For comparison, Ueno et al. [68] reported temperature lapse rates in the mountainous region of central Japan, which includes the steep river basin targeted in this study, of 0.59˚C (100 m)$^{-1}$ during March–May, 0.55˚C (100 m)$^{-1}$ during June–

August, 0.55°C $(100 \text{ m})^{-1}$ during September–November, and 0.58°C $(100 \text{ m})^{-1}$ during December–February. In contrast, studies in Europe indicated that the temperature lapse rates in the Pyrenees and Slovakia were only 0.44–0.51°C $(100 \text{ m})^{-1}$ in 2005 and 2006 [11], but they averaged 0.59°C $(100 \text{ m})^{-1}$ between 1981 and 2010 [23].

To assess the accuracy of the temperature lapse rate of 0.6°C $(100 \text{ m})^{-1}$, we examined the differences between the observed daily mean temperature at TKY-weather station (1342 m a.s. l.) and that estimated from the observed daily mean temperature at Takayama-AMeDAS (560 m a.s.l.) and a temperature lapse rate of 0.6°C $(100 \text{ m})^{-1}$. The differences displayed a seasonal pattern (Fig 9; Table 5). The differences (estimated – observed) were −0.62°C during January–March and −1.25°C during October–December. The temperatures estimated with a lapse rate

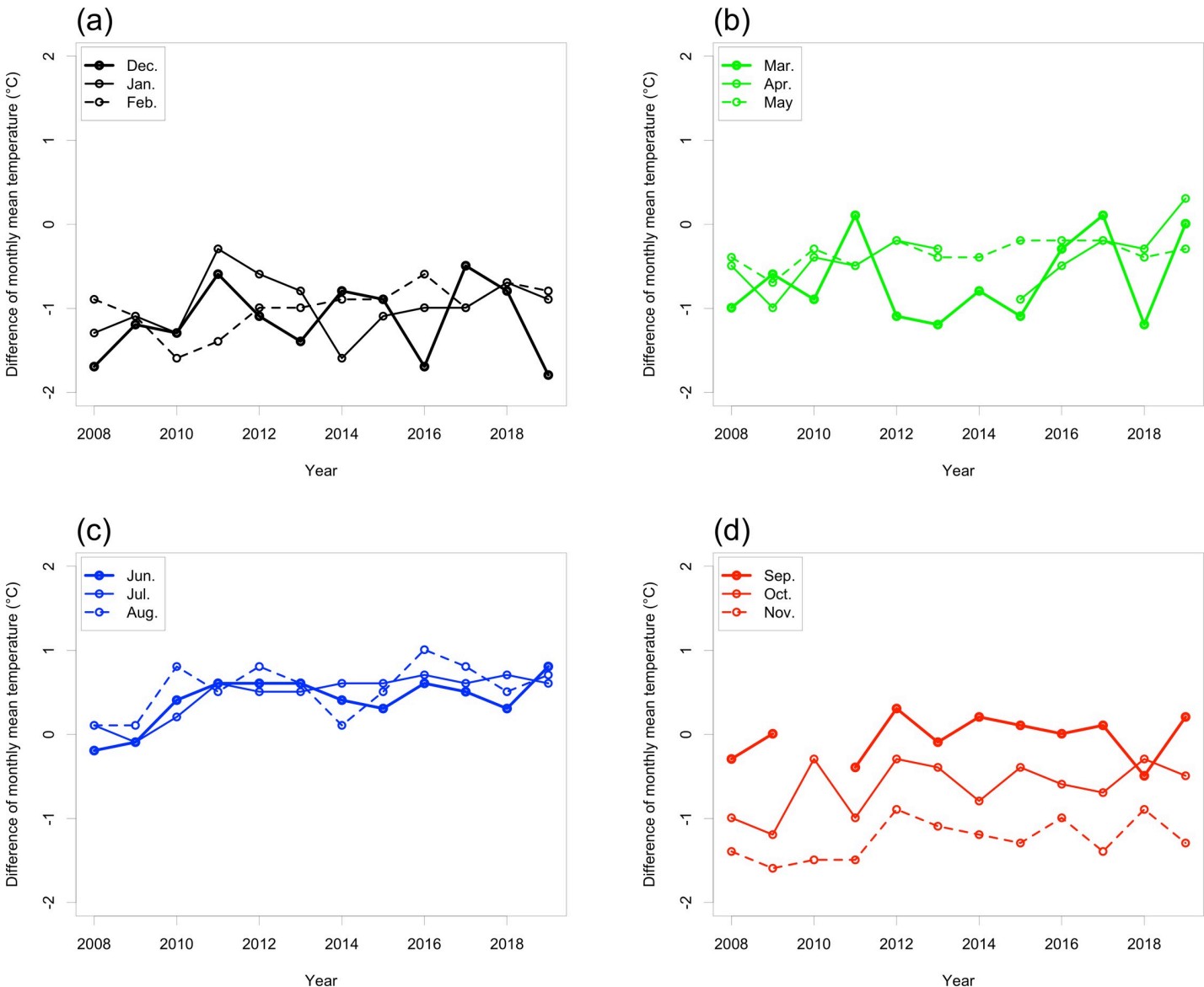

**Fig 9. Summary of differences at TKY-weather station (1342 m a.s.l.) between observed daily mean temperature and that estimated by using the temperature observed at TKY-AMeDAS (560 m a.s.l.) (estimated minus observed) and a temperature lapse rate of 0.6°C $(100 \text{ m})^{-1}$.** (a) Dec, Jan, Feb; (b) Mar, Apr, May: (c) Jun, Jul, Aug; and (d) Sep, Oct, Nov. See statistics in Table 5.

**Table 5. Summary of differences between temperatures (˚C) observed at TKY-weather station (1342 m a.s.l.) and those estimated by using daily mean temperatures at Takayama-AMeDAS (560 m a.s.l.) (estimated minus observed) and a temperature lapse rate of 0.6˚C (100 m)$^{-1}$.**

| Month | Ave. | S.D. | Max. | Min. |
|---|---|---|---|---|
| Jan. | −0.97 | 0.35 | −0.29 | −1.59 |
| Feb. | −0.98 | 0.28 | −0.59 | −1.59 |
| Mar. | −0.66 | 0.51 | 0.11 | −1.19 |
| Apr. | −0.40 | 0.35 | 0.31 | −0.99 |
| May | −0.34 | 0.15 | −0.19 | −0.69 |
| Jun. | 0.41 | 0.3 | 0.81 | −0.19 |
| Jul. | 0.47 | 0.26 | 0.71 | −0.09 |
| Aug. | 0.55 | 0.31 | 1.01 | 0.11 |
| Sep. | −0.03 | 0.26 | 0.31 | −0.49 |
| Oct. | −0.62 | 0.31 | −0.29 | −1.19 |
| Nov. | −1.25 | 0.24 | −0.89 | −1.59 |
| Dec. | −1.14 | 0.44 | −0.49 | −1.79 |
| All | −0.42 | 0.69 | 1.01 | −1.79 |

of 0.6˚C (100 m)$^{-1}$ therefore underestimated the observed temperatures at TKY-weather station. However, the differences in April, May, and September, which are important times to calculate SGS and EGS dates, were comparatively small and ranged from −0.03 to −0.40˚C. Furthermore, the differences between June and August were positive and ranged from 0.41 to 0.55˚C. The seasonality of the temperature lapse rates is primarily a result of the seasonality of humidity [30,68–70]. The small differences between the estimated and observed daily mean temperatures (0.03–0.40˚C) in April, May, and September may have caused errors of a few days in the modelled SGS and EGS dates. Consideration should also be given to the effect of temperature inversions on the SGS date in a river basin [30].

These observations suggest that the temperature lapse rate of 0.6˚C (100 m)$^{-1}$ may slightly underestimate the temperature in high-elevation areas, except during summer. However, images of phenology taken on modelled SGS and EGS dates in each year at TKY-tower showed almost the same condition of the canopy surface, although there were small variations in leaf colouring and leaf fall (Fig 5). This result suggests that interannual variability of SGS and EGS dates at a site can be evaluated in a relative sense with a constant lapse rate. Important tasks for the future will include meteorological observations in mountainous regions (i.e., greater than 1000 m a.s.l.) where there are few weather stations such as the AMeDAS [13] and examination of the temperature lapse rate in each river basin.

## 4.4. Further acquisition of ground truth data

We used phenology and live camera images taken in the upper and lower reaches of the Daihachiga River basin as validation data (Figs 5 and 6). However, evaluation of the uncertainty of the spatial representation of phenological events by our model has been constrained by the amount of data available to us. The development of degree-day models of phenology to estimate SGS and EGS dates will require many ground truth observations of phenology at multiple points along an elevational gradient. In particular, reducing the uncertainties in the areal coverages of different tree species captured in phenological and live-camera images will require many observations at multiple points. We have recently been able to easily acquire live camera images taken at multiple tourist sites and riversides. Silva et al. [71] reported that phenological information can be extracted from analysis of images uploaded on Twitter. However, those

images are poorly suited for detecting plant phenology if the percentage of vegetation in them is low, and the purpose of capturing the images is completely different (e.g., weather monitoring, tourism, disaster prevention, and recreation). Such images may therefore be difficult to use for quantitative analyses (e.g., extraction of RGB values from daily images of phenology; e.g., [31,72,73]). In addition, it is difficult to obtain such images in forests and mountainous regions, where there are few requirements related to traffic and societal needs (e.g., tourism and disaster prevention).

Websites on the Internet that can be easily accessed for uploading images with geotags (e.g., Mapillary at https://www.mapillary.com, accessed 6 July 2021) can be used to upload and download images with properties such as GPS (Global Positioning System) coordinates, date, time, and contributor. This capability suggests that images taken along roads within a river basin and periodically archived could be used to evaluate the spatiotemporal variability of SGS and EGS dates. In fact, Ishihara et al. [29] used daily movie-camera images taken along roads in the Daihachiga River basin to document phenology. One of us (K.N.N.) has already used a drive recorder to capture images of the Daihachiga River basin and has uploaded the images with GPS logs to Mapillary. Important tasks for the future include comparing periodically archived ground-truth images, data recorded by SENTINEL-2A/2B-satellites, and modelled SGS and EGS dates.

## 5. Conclusions

We used a degree-day phenology model to evaluate the spatiotemporal variability of the SGS and EGS dates in the Daihachiga River Basin on a century timescale and the relationship of that variability to changes of temperature. We found that (1) the sensitivity of the modelled SGS and EGS dates to elevation changed from 3.29 days $(100\,\mathrm{m})^{-1}$ ($-5.48$ days $°C^{-1}$) and $-2.89$ days $(100\,\mathrm{m})^{-1}$ (4.81 days $°C^{-1}$), respectively, in 1900 to 2.85 days $(100\,\mathrm{m})^{-1}$ ($-4.75$ days $°C^{-1}$) and $-2.84$ day $(100\,\mathrm{m})^{-1}$ (4.73 day $°C^{-1}$), respectively, in 2019, and (2) the long-term trend of the sensitivity of the modelled SGS date to elevation was $-0.0037$ day year$^{-1}$ per 100 m, but the analogous trend in the case of the modelled EGS date was not significant. Despite the use of daily phenology and live camera images as well as satellite data with a high spatial resolution to assess the generality and representativeness of the modelled SGS and EGS dates, limitations of our study still remain. The next steps will require (1) examination of the relationship between elevation and SGS and EGS dates in other steep river basins where annual mean temperatures are much higher than the temperatures in the Daihachiga River basin, (2) integrative evaluation of the SGS and EGS dates based on a degree-day phenological model and satellite observations with a fine spatial resolution and high observation frequency, (3) acquisition of an accurate basin-specific temperature lapse rate in each target basin, and (4) further acquisition of ground truth data such as live camera images and images with geotags on the Internet. Plant phenology was more sensitive to elevation than to latitude in a local area (e.g., within 2° latitude × 3° longitude) [9], where we could ignore the effect of phenological plasticity associated with climate change. Hence, the development of degree-day phenological models in multiple steep river basins with different climate conditions will deepen our ecological understanding of the sensitivity of spring and autumn phenology to future climate change.

## Acknowledgments

We thank the Koito-Pottery and Takayama Printing Co, Ltd, for providing live camera images. We thank all members of the Takayama community for their assistance in the field. We thank the editor and anonymous reviewers for their constructive comments.

## Author Contributions

**Conceptualization:** Nagai Shin, Taku M. Saitoh.

**Formal analysis:** Nagai Shin.

**Funding acquisition:** Nagai Shin.

**Methodology:** Nagai Shin, Taku M. Saitoh, Kenlo Nishida Nasahara.

**Supervision:** Nagai Shin, Taku M. Saitoh, Kenlo Nishida Nasahara.

**Visualization:** Nagai Shin.

**Writing – original draft:** Nagai Shin.

**Writing – review & editing:** Nagai Shin, Taku M. Saitoh, Kenlo Nishida Nasahara.

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
