## [Decision Letter · Decision Letter 0]

31 Dec 2020

PONE-D-20-20028

How did the characteristics of the growing season change during the past 100 years at a steep river basin in Japan?

PLOS ONE

Dear Dr. Nagai,

Thank you for submitting your manuscript to PLOS ONE. After careful consideration, we feel that it has merit but does not fully meet PLOS ONE’s publication criteria as it currently stands. Therefore, we invite you to submit a revised version of the manuscript that addresses the points raised during the review process.

Please pay attention to the discussion of the causes of altitudinal differences in phenological patterns.

We look forward to receiving your revised manuscript.

Kind regards,

Dusan Gomory

Academic Editor

PLOS ONE

Journal Requirements:

2.We note that [Figure(s) 1, 7 and 8] in your submission contain map/satellite images which may be copyrighted. All PLOS content is published under the Creative Commons Attribution License (CC BY 4.0), which means that the manuscript, images, and Supporting Information files will be freely available online, and any third party is permitted to access, download, copy, distribute, and use these materials in any way, even commercially, with proper attribution. For these reasons, we cannot publish previously copyrighted maps or satellite images created using proprietary data, such as Google software (Google Maps, Street View, and Earth). For more information, see our copyright guidelines: http://journals.plos.org/plosone/s/licenses-and-copyright.

1.    You may seek permission from the original copyright holder of Figure(s) [1, 7 and 8] to publish the content specifically under the CC BY 4.0 license. 

Additional Editor Comments (if provided):

I apologize for excessive time needed to get the reviews, but...

Reviewers' comments:

Reviewer's Responses to Questions

**Comments to the Author**

1. Is the manuscript technically sound, and do the data support the conclusions?

Reviewer #1: Yes

Reviewer #2: Yes

2. Has the statistical analysis been performed appropriately and rigorously? 

Reviewer #1: Yes

Reviewer #2: Yes

3. Have the authors made all data underlying the findings in their manuscript fully available?

Reviewer #1: Yes

Reviewer #2: Yes

4. Is the manuscript presented in an intelligible fashion and written in standard English?

Reviewer #1: Yes

Reviewer #2: No

5. Review Comments to the Author

Reviewer #1: This paper deals with calculation of start and end of the growing season in a steep river basin located in a mountainous region of central Japan under ongoing climate change. There not many studies have been done on this topic, so I really appreciate, that authors decided to solve this interesting topic and found literature. They chose the appropriate procedures, and it is admirable that they connected camera images and satellite data as well. This a future of many research topics.

I appreciate that the presented paper addresses a very important topic, it is to know that the authors have read a large amount of similar literature. But while the paper appears to be sound, the language is unclear, there are too many numbers in the text making it difficult to follow (maybe it is not necessary to state all numbers). I advise the authors work with a writing coach or copyeditor to improve the flow and readability of the text (e.g. do not use in text form such as „we“or similar).

And the most important point - I am missing very important part of the paper – CONCLUSION – please add it..

Some references are missing in the text (e.g. 2, 3, 4, 41, 42, 53, 58, 59, 60).

Furthermore, there are few typos in the text e.g. line 158 < 375.1 °C, better to check the whole text one more time.

Figure 9 is a little bit unclear, too many lines – I suggest to adjust it (maybe divide into 2 charts).

After adjustment, I recommend publishing this reseach paper.

Reviewer #2: This study addresses change of plant phenological events due to climate change. Authors emphasize that the effects of climate change on the phenological events may be greater in a steep river basin than at the scales of the nation and continent. They calculated the start (SGS) and end (EGS) dates of the growing season and thus compared the dates between a steep river basin (1420 m a.s.l.) and meteorological station (560 m a.s.l.). Furthermore, they also obtained the green-red vegetation index from satellite image and thus compared the index between both sites.

But there are some questions. First of all, there is a question about Table 3's data. The difference in altitude between the two sites is 860 m. Then the temperature difference according to the altitude should be about 5°C, but there is not that much difference.

Urbanization is known to have a greater impact than a rise in carbon dioxide as a cause of climate change. However, the results of this study indicate that changes in phenological events are occurring faster in site located on the higher altitudes than in the site of lower elevation with high land-use intensity. The authors estimated the background as follows: This possibility is due to the effect of temperature on plant phenology and the difference between vertical and horizontal gradients in temperature sensitivities. But there is no detailed evidence. An explanation of this part should be at the heart of this paper.

It is very difficult to read the data. If the date calculated by the measured temperature and the date obtained by analyzing the satellite image data are tabulated, it will be easier to read the data.

In addition, it would also be meaningful to obtain changes in the spatial distribution of temperature from satellite imaging data and compare them with the results of the GVI.

Finally, the paper could benefit from some editing for grammar and usage in some places—syntax is at times awkward.

6. PLOS authors have the option to publish the peer review history of their article (what does this mean?). If published, this will include your full peer review and any attached files.

Reviewer #1: No

Reviewer #2: **Yes: **Chang Seok Lee

---

## [Author Response · Author response to Decision Letter 0]

16 Mar 2021

Dr. Dusan Gomory

Academic Editor of PLOS ONE

Dear Dr. Dusan Gomory:

We sincerely appreciate your coordination of the review and comments on the following submitted paper:

ID: PONE-D-20-20028

Title: How did the characteristics of the growing season change during the past 100 years at a steep river basin in Japan?

We have substantially revised the manuscript in accordance with the review comments. In addition, we have also re-checked all data and revised some minor points (including tables and figures). The portions of the manuscript that we revised are coloured red. Our detailed responses to the comments are provided in this letter. We hope that our responses and the resulting changes will be satisfactory, but we will be happy to work with you to resolve any remaining issues.

 We request permission for the open-access journal PLOS ONE to publish Shin et al. under the Creative Commons Attribution License (CCAL) CC BY 4.0 (http://creativecommons.org/licenses/by/4.0/). Please be aware that this license allows unrestricted use and distribution, even commercially, by third parties. Please reply and provide explicit written permission to publish Shin et al. under a CC BY license and complete the attached form.

Sincerely,

Shin Nagai (on behalf of all authors)

Response to reviewers’ comments:

Academic editor: 

Please pay attention to the discussion of the causes of altitudinal differences in phenological patterns.

Answer:

Thank you for your efforts to improve our paper. We have added the following discussion (lines 307−336). 

The sensitivity of the modelled SGS and EGS dates to elevation changed from 3.29 days (100 m)−1 (−5.48 day °C−1) and −2.89 days (100 m)−1 (4.81 day °C−1), respectively, in 1900 to 2.85 days (100 m)−1 (−4.75 day °C−1) and −2.84 days (100 m)−1 (4.73 day °C−1) in 2019 (Fig. 2). The implication is that the modelled SGS date at Takayama-AMeDAS (560 m a.s.l.) in 1900 (DOY 124) was equivalent to the SGS date at 843 m a.s.l. in 2020 (shift to 283 m higher elevation). The modelled EGS date at Takayama-AMeDAS (560 m a.s.l.) in 1900 (DOY 317) was equivalent to the EGS at 861 m a.s.l. in 2019 (shift to 301 m higher elevation). However, the yearly difference of the modelled SGS dates at Takayama-AMeDAS and TKY ranged from −15 days (earlier than previous year) to 21 days (later than previous year) and from −16 days to 23 days, respectively (Fig. 2a). These differences are equivalent to elevation changes of −526 m (shift to lower elevation) to +737 m (shift to higher elevation) and to elevation changes of −561 m to +807 m, respectively (in the case of 2019). The yearly difference of the modelled EGS dates at Takayama-AMeDAS and TKY ranged from −10 days (earlier than previous year) to 17 days (later than previous year) and from −12 days to 15 days, respectively (Fig. 2b). These differences were equivalent to elevation changes of +352 m to −599 m and +423 m to −528 m, respectively (in the case of 2019). These comparisons suggest that deciduous, broad-leaved forests in the Daihachiga River basin might have acclimated to the year-to-year variability of climate during the past 120 years by changing their leaf flush and leaf fall phenology. In other words, the warmer spring temperature triggered earlier SGS dates, and the warmer autumn temperature triggered later EGS dates. However, this acclimation may not apply to other river basins where annual mean temperatures are much higher than the temperatures of the Daihachiga River basin. Nagai et al. (2020) [57] reported that the correlation between the observed first flowering date of Yoshino cherry and latitude has decreased since 1980 in Japan. They explained the lower correlation by reasoning that higher temperatures in winter had delayed the release from dormancy and the subsequent first-flowering dates in areas where the annual mean temperature was high. To evaluate the significance of the chilling requirement for dormancy release, we will need to examine the relationships between elevation and the dates of SGS and EGS in other steep river basins where annual mean temperatures are much higher than the temperatures in the Daihachiga River basin.

Reviewer 1: 

This paper deals with calculation of start and end of the growing season in a steep river basin located in a mountainous region of central Japan under ongoing climate change. There not many studies have been done on this topic, so I really appreciate, that authors decided to solve this interesting topic and found literature. They chose the appropriate procedures, and it is admirable that they connected camera images and satellite data as well. This a future of many research topics.

Answer: 

Thank you for your efforts to improve our paper. In accordance with your kind and constructive comments, we have revised the manuscript. In addition, we have also re-checked all data and revised some minor points (including tables and figures).

Comment 1: I appreciate that the presented paper addresses a very important topic, it is to know that the authors have read a large amount of similar literature. But while the paper appears to be sound, the language is unclear, there are too many numbers in the text making it difficult to follow (maybe it is not necessary to state all numbers). I advise the authors work with a writing coach or copyeditor to improve the flow and readability of the text (e.g. do not use in text form such as „we“or similar). 

Answer: 

The English in this manuscript was carefully corrected by two native-English speaking, professional editors, both with extensive research editing experience.

Comment 2: And the most important point - I am missing very important part of the paper – CONCLUSION – please add it.. 

Answer:

We have added the following “Conclusion” section.

5. Conclusion

We used a degree-day phenological model to successfully quantify the sensitivity of the modelled SGS and EGS dates to elevation in the Daihachiga River basin on a century timescale. The next steps will require (1) examination of the relationship between elevation and SGS and EGS dates in other steep river basins where annual mean temperatures are much higher than the temperatures in the Daihachiga River basin, (2) integrative evaluation of the SGS and EGS dates based on a degree-day phenological model and satellite observations with a fine spatial resolution and high observation frequency, (3) acquisition of an accurate basin-specific temperature lapse rate in each target basin, and (4) further acquisition of ground truth data such as live camera images and images with geotags on the Internet. The development of degree-day phenological models in multiple steep river basins with different climate conditions will deepen our ecological understanding of the sensitivity of spring and autumn phenology to future climate change.

Comment 2: Some references are missing in the text (e.g. 2, 3, 4, 41, 42, 53, 58, 59, 60). 

Answer: 

We have checked the list of references.

Comment 3: Furthermore, there are few typos in the text e.g. line 158 < 375.1 °C, better to check the whole text one more time. 

Answer: 

The English in this manuscript was carefully corrected by two native-English speaking, professional editors, both with extensive research editing experience.

Comment 4: Figure 9 is a little bit unclear, too many lines – I suggest to adjust it (maybe divide into 2 charts). 

Answer: 

We have divided into 4 charts. Please see Figure 9.

Reviewer 2: 

This study addresses change of plant phenological events due to climate change. Authors emphasize that the effects of climate change on the phenological events may be greater in a steep river basin than at the scales of the nation and continent. They calculated the start (SGS) and end (EGS) dates of the growing season and thus compared the dates between a steep river basin (1420 m a.s.l.) and meteorological station (560 m a.s.l.). Furthermore, they also obtained the green-red vegetation index from satellite image and thus compared the index between both sites. 

Answer: 

Thank you for your efforts to improve our paper. In accordance with your kind and constructive comments, we have revised the manuscript. In addition, we have also re-checked all data and revised some minor points (including tables and figures).

Comment 1: But there are some questions. First of all, there is a question about Table 3’s data. The difference in altitude between the two sites is 860 m. Then the temperature difference according to the altitude should be about 5°C, but there is not that much difference. 

Answer: 

You might be misled. In table 3, to validate the temperature lapse rate of 0.6 °C (100 m)−1, we examined the differences between temperatures observed at TKY-weather station (1342 m a.s.l.) and those estimated by using daily mean temperatures at Takayama-AMeDAS (560 m a.s.l.) (estimated minus observed) and a temperature lapse rate of 0.6 °C (100 m)-1. The differences displayed a seasonal pattern (Fig. 9; Table 4). The seasonality of the temperature lapse rates is primarily a result of the seasonality of humidity [30, 62–64]. The small differences between the estimated and observed daily mean temperatures (0.03–0.40 °C) in April, May, and September may have caused errors of a few days in the modelled SGS and EGS dates.

Comment 2: Urbanization is known to have a greater impact than a rise in carbon dioxide as a cause of climate change. However, the results of this study indicate that changes in phenological events are occurring faster in site located on the higher altitudes than in the site of lower elevation with high land-use intensity. The authors estimated the background as follows: This possibility is due to the effect of temperature on plant phenology and the difference between vertical and horizontal gradients in temperature sensitivities. But there is no detailed evidence. An explanation of this part should be at the heart of this paper. 

Answer: 

First, we agree with your comment. The effect of heat island due to urbanization should be deliberated for long-term time scale, especially in a big city. However, Takayama is a small city whose population is about 90,000. We think it’s not so easy to distinguish between the effect of heat island and natural climate change clearly. Therefore, we assumed that the heat island effect on temperature caused by urbanization was relatively low at Takayama-AMeDAS. Despite merging of municipalities in 2005, the population of the city of Takayama increased from 63,520 in 1920 to only 89,182 in 2015 (https://www.city.takayama.lg.jp/_res/projects/default_project/_page_/001/011/771/18gou_siryou2.pdf, accessed 9 March 2021). We have mentioned this in the “Method” section (lines 153−158).

Second, to answer for your second comment, we have added the following discussion (lines 307−336). 

The sensitivity of the modelled SGS and EGS dates to elevation changed from 3.29 days (100 m)−1 (−5.48 day °C−1) and −2.89 days (100 m)−1 (4.81 day °C−1), respectively, in 1900 to 2.85 days (100 m)−1 (−4.75 day °C−1) and −2.84 days (100 m)−1 (4.73 day °C−1) in 2019 (Fig. 2). The implication is that the modelled SGS date at Takayama-AMeDAS (560 m a.s.l.) in 1900 (DOY 124) was equivalent to the SGS date at 843 m a.s.l. in 2020 (shift to 283 m higher elevation). The modelled EGS date at Takayama-AMeDAS (560 m a.s.l.) in 1900 (DOY 317) was equivalent to the EGS at 861 m a.s.l. in 2019 (shift to 301 m higher elevation). However, the yearly difference of the modelled SGS dates at Takayama-AMeDAS and TKY ranged from −15 days (earlier than previous year) to 21 days (later than previous year) and from −16 days to 23 days, respectively (Fig. 2a). These differences are equivalent to elevation changes of −526 m (shift to lower elevation) to +737 m (shift to higher elevation) and to elevation changes of −561 m to +807 m, respectively (in the case of 2019). The yearly difference of the modelled EGS dates at Takayama-AMeDAS and TKY ranged from −10 days (earlier than previous year) to 17 days (later than previous year) and from −12 days to 15 days, respectively (Fig. 2b). These differences were equivalent to elevation changes of +352 m to −599 m and +423 m to −528 m, respectively (in the case of 2019). These comparisons suggest that deciduous, broad-leaved forests in the Daihachiga River basin might have acclimated to the year-to-year variability of climate during the past 120 years by changing their leaf flush and leaf fall phenology. In other words, the warmer spring temperature triggered earlier SGS dates, and the warmer autumn temperature triggered later EGS dates. However, this acclimation may not apply to other river basins where annual mean temperatures are much higher than the temperatures of the Daihachiga River basin. Nagai et al. (2020) [57] reported that the correlation between the observed first flowering date of Yoshino cherry and latitude has decreased since 1980 in Japan. They explained the lower correlation by reasoning that higher temperatures in winter had delayed the release from dormancy and the subsequent first-flowering dates in areas where the annual mean temperature was high. To evaluate the significance of the chilling requirement for dormancy release, we will need to examine the relationships between elevation and the dates of SGS and EGS in other steep river basins where annual mean temperatures are much higher than the temperatures in the Daihachiga River basin.

Comment 3: It is very difficult to read the data. If the date calculated by the measured temperature and the date obtained by analyzing the satellite image data are tabulated, it will be easier to read the data. 

Answer: 

We have added the following table. 

Table 2. Summary of relationships between the modelled SGS and EGS dates, and satellite data.

Phenology Location Modelled Satellite Difference (modelled-satellite)

SGS Takayama-AMeDAS 18 Apr. 2018 (DOY 108) 21 Apr. 2018 (DOY 111) −3

 TKY-tower 14 May 2018 (DOY 134) 16 May 2018 (DOY 136) −2

EGS Takayama-AMeDAS 25 Nov. 2018 (DOY 329) 27 Nov. 2018 (DOY 331) −2

 TKY-tower 30 Oct. 2018 (DOY 303) 2 Nov. 2018 (DOY 306) −3

Comment 4: In addition, it would also be meaningful to obtain changes in the spatial distribution of temperature from satellite imaging data and compare them with the results of the GVI. 

Answer: 

You might be misled. We did not evaluate the spatial distribution of temperature from satellite data. 

Comment 5:

Finally, the paper could benefit from some editing for grammar and usage in some places—syntax is at times awkward. 

Answer: 

The English in this manuscript was carefully corrected by two native-English speaking, professional editors, both with extensive research editing experience.

---

## [Decision Letter · Decision Letter 1]

10 Jun 2021

PONE-D-20-20028R1

How did the characteristics of the growing season change during the past 100 years at a steep river basin in Japan?

PLOS ONE

Dear Dr. Nagai,

Thank you for submitting your manuscript to PLOS ONE. After careful consideration, we feel that it has merit but does not fully meet PLOS ONE’s publication criteria as it currently stands. Therefore, we invite you to submit a revised version of the manuscript that addresses the points raised during the review process.

Please, pay attention to reviewers' comments, especially those related to conclusions and section 3.2.

We look forward to receiving your revised manuscript.

Kind regards,

Dusan Gomory

Academic Editor

PLOS ONE

Journal Requirements:

Reviewers' comments:

Reviewer's Responses to Questions

**Comments to the Author**

1. If the authors have adequately addressed your comments raised in a previous round of review and you feel that this manuscript is now acceptable for publication, you may indicate that here to bypass the “Comments to the Author” section, enter your conflict of interest statement in the “Confidential to Editor” section, and submit your "Accept" recommendation.

Reviewer #3: (No Response)

Reviewer #4: All comments have been addressed

2. Is the manuscript technically sound, and do the data support the conclusions?

Reviewer #3: Partly

Reviewer #4: Partly

3. Has the statistical analysis been performed appropriately and rigorously? 

Reviewer #3: No

Reviewer #4: Yes

4. Have the authors made all data underlying the findings in their manuscript fully available?

Reviewer #3: (No Response)

Reviewer #4: Yes

5. Is the manuscript presented in an intelligible fashion and written in standard English?

Reviewer #3: (No Response)

Reviewer #4: No

6. Review Comments to the Author

Reviewer #3: Abstract

I´m missing 2-3 sentences at the end of abstract about meaningful of this study. Why authors did this study and what they wanted to say.

Introduction

From line 97 (…“we carried….“) to line 107 – this is methodology and should be part of methodology chapter. Maybe hypothesis should be listed here.

Material and methods

I´m missing information about phenology, phenological phases, species which were observed by cameras or personally. There are only small information about location and species in 2.1. In 2.4. there are information about camreas and twoo localities but I can not see information about phenological phases which were observed, nor species.

Results

In 3.2 – whole chapter is confusing – all data could be seen in picture or much better they could be in table. It is not necessary to write out all values and changes of standard deviations. I also do not understand why the changes of SD in different decades are important for reader (lines 237-239). This chapter should be rewrited.

In 3.4 – there should not be used the expression „validation“ – there are no data which were validated, there is only information about lets say comparison of modelled values with pictures from phenological cameras. If you want to do validation it is necessary to use some statistical indicators (e.g. RMSE) for calculating the terms of the start of growing season and end of GS. Only compariosn with picutes are not enough for validation.

In 3.4 – lines 258 and 261 – this is discussion not results.

In 3.5 – first sentence – it is not clear for me, I can not understand what the authors wanted to say. It should be maybe comparison of modelled data of start and end of GS with satellite data. Anyway the text is somehow heavy.

In 3.5 – I´m wondering if the satellite data should be used within this study at all. Information from only one year can not say enough.

Discussion

In 4.1 – first sentence – it is relust not part for discussion. Lines 290-291 – not clear why authors compared their results with other species when we do not know which species were used by authors. Maybe the comparison with location or simply with other papers should be better.

Lines 311-312. If I understand well the authors said that SGS and EGS dates defined by satellite only in one year 2018 differed from the dates observed in-situ. I´m not sure if data from only one year are suitable for such conlcusion.

In 4.4 – lines 385-391 – this is rather methodology but absolutely not the discussion.

4.4 – the importance of this chapter is unclear for me. Maybe it is important but there is no explanation by authors why this is important for the whole study. This should be excluded or rewrite.

Conclusion – I´m missing conclusions and infromation is authors fulfil and answer the goals of the study. I´m not sure what authors wanted to say by this study at all.

Reviewer #4: The manuscript entitled ‘How did the characteristics of the growing season change during the past 100 years at a steep river basin in Japan?’ by Nagai et al deals with the growing season in a mountainous region of Japan. Paper is scientific but needs improvement. My observations on the follow up correspondence are as follows:

Comments

Does number of sunny days play any role in the higher elevation to the phenology? I do agree with the author about the urban heat island explanation but want to see if temperature gradient with available sunny days have any impact needs to be studied.

Use of the punctuation marks and conjunction e.g. ‘and’ in the abstract and introduction section need revision [e.g. L-27, 68-69] due to unnecessary use at times.

The North arrow is an important component of the map and therefore it must be incorporated in each map.

What are the limitations of the study? Elaboration a little on this aspect of the study.

L- 210 should be “We used RGB composite images and calculated the green-red vegetation index (GRVI; [54]).”

Mention in the article, if any cloud removal method used in the research study to get accurate information from the SENTINEL data.

Used acronyms need to be in their full form the first time they are used. This issue is consistent in the whole manuscript.

The method used to calculate the trend, correlation result, and significance of the results need to be elaborated. The level of significance used to consider the result ‘significant’ should also be mentioned.

Camera images and satellite observation were used to validate the obtained results. Therefore, it is better to make one common section 3.4 for validation through camera image and satellite observation. Otherwise, section 3.4 can be split into two sub-sections i.e. one for camera images and the other one for satellite observation.

The conclusions section is very weak. It should provide a summary of the whole study, including major results of the study, method, issue addressed.

7. PLOS authors have the option to publish the peer review history of their article (what does this mean?). If published, this will include your full peer review and any attached files.

Reviewer #3: No

Reviewer #4: No

---

## [Author Response · Author response to Decision Letter 1]

7 Jul 2021

Dr. Dusan Gomory

Academic Editor of PLOS ONE

Dear Dr. Dusan Gomory:

We sincerely appreciate your coordination of the review and comments on the following submitted paper:

ID: PONE-D-20-20028

Title: How did the characteristics of the growing season change during the past 100 years at a steep river basin in Japan?

We have substantially revised the manuscript in accordance with the review comments. The portions of the manuscript that we revised are coloured red. Our detailed responses to the comments are provided in this letter. We hope that our responses and the resulting changes will be satisfactory, but we will be happy to work with you to resolve any remaining issues.

Sincerely,

Shin Nagai (on behalf of all authors)

Response to reviewers’ comments:

Reviewer 3: 

Comment 1: Abstract

I’m missing 2-3 sentences at the end of abstract about meaningful of this study. Why authors did this study and what they wanted to say. 

Answer: 

Thank you for your efforts to improve our paper. In accordance with your kind and constructive comments, we have revised the manuscript. At the end of “Abstract”, we have revised as follows (lines 37−40).

“Despite the need for further studies to improve the generality and representativeness of the model, the development of degree-day phenology models in multiple, steep river basins will deepen our ecological understanding of the sensitivity of plant phenology to climate change.”

Comment 2: Introduction

From line 97 (…“we carried….“) to line 107 – this is methodology and should be part of methodology chapter. Maybe hypothesis should be listed here. 

Answer: 

At the end of “Introduction” section, we have revised as follows (lines 99−108).

“Accordingly, we calculated the SGS and EGS dates from 1900 to 2020 by using a degree-day phenological model that was developed for a cool-temperate, deciduous, broad-leaved forest site located in the upper reaches of the basin [32]. We then assessed the accuracy of the modelled dates by using daily images of vegetation phenology taken in the upper reaches of the basin [39, 40], daily live camera images taken in the lower reaches [41], and satellite observations with a spatial resolution of 10 m. The goals of this study were (1) to examine the characteristics of the spatiotemporal variability of the SGS and EGS dates in a steep river basin under the influence of climate change on a century timescale and to identify what caused the variability, and (2) to assess the generality and representativeness of a degree-day model of vegetation phenology.

”

Comment 3: Material and methods

I’m missing information about phenology, phenological phases, species which were observed by cameras or personally. There are only small information about location and species in 2.1. In 2.4. there are information about cameras and two localities but I cannot see information about phenological phases which were observed, nor species.

Answer: 

In the “Material and methods”, and “Results” sections, we have described the information of tree species and observed by cameras as follows. 

Material and methods section (lines 130−142): 

“The typical landscape within and around the Daihachiga River basin consists of deciduous, broad-leaved forests (mainly deciduous oak and birch), deciduous, coniferous forests (larch), and evergreen, coniferous forests (mainly Japanese cedar and cypress). Deciduous, broad-leaved forests are distributed mainly on the southern slope at elevations of 800–1400 m a.s.l. [48]. At TKY, the dominant canopy tree species are Quercus crispula (deciduous oak) and Betula ermanii (birch), with some Fagus crenata (beech), Betula platyphylla var. japonica (birch), and Magnolia obovata (magnolia). The sub-dominant canopy tree species are Acer distylum (lime-leaved maple), Acer rufinerve (snakebark maple), Acanthopanax sciadophylloides, Tilia japonica (Japanese linden), Sorbus alnifolia (Korean mountain ash), and Kalopanax pictus (castor aralia). The dominant shrub tree species are Hydrangea paniculata (panicled hydrangea) and Viburnum furcatum (forked viburnum). The forest floor is fully covered by an evergreen dwarf bamboo (Sasa senanensis; striped bamboo) [49, 50].” 

“Results” section (lines 209−229): 

“To assess the generality of the modelled SGS and EGS dates, we used daily images of phenology taken at TKY-tower [39, 40] and live camera images taken at two points near Takayama-AMeDAS: Kaji-bridge, 1.5 km south (36°8′367″N, 137°15′27″E, 572 m a.s.l.; Figs. 7, 8; [43]), and Koito-Pottery, 2.6 km north-east (36°8′8″N, 137°14′24″E, 605 m a.s.l.; Figs. 7, 8, [43]) of Takayama-AMeDAS (Nagai et al., 2018b). At TKY-tower, daily images of phenology have been publicly available since 2003 on the web site of the Phenological Eyes Network (http://www.pheno-eye.org, accessed 6 July 2021). The daily images of phenology showed mainly Q. crispula, B. ermanii (dominant canopy tree species), A. distylum (sub-dominant canopy tree species), and Prunus maximowiczii [58, 59]. At Koito-Pottery and Kaji-bridge, the latest live camera images are publicly available on an hourly or 3-minute timeframe from the Web sites of the Koito Pottery (https://koitoyaki.com/livecam/, which we have not accessed since March 2021) and Takayama Printing Co, Ltd (https://www.takayama-dp.com/live/, accessed 6 July 2021). At Koito-Pottery, the live camera images showed mainly deciduous trees close to the cameras and distant deciduous and evergreen forests. At Kaji-bridge, the live camera images showed mainly deciduous trees close to the cameras and distant deciduous forests. We assumed that there was no difference between the SGS and EGS dates at Takayama-AMeDAS, Koito-Pottery, and Kaji-bridge, because the difference in elevation among the three sites is very small (max. 45 m). We have downloaded live camera images by running shell scripts and the crontab command on a Linux personal computer since December 2017. We obtained permission to download images from the Koito Pottery and Takayama Printing Co, Ltd.”

Comment 4: Results

In 3.2 – whole chapter is confusing – all data could be seen in picture or much better they could be in table. It is not necessary to write out all values and changes of standard deviations. I also do not understand why the changes of SD in different decades are important for reader (lines 237-239). This chapter should be rewritten.

Answer: 

In the “Material and methods” section, we have added descriptions to explain the importance of SD. We have also revised the section 3.2. 

“Material and methods: 2.2. daily mean air temperature” section (lines 168−174):

“To evaluate the characteristics of climate change, we examined the timeseries of 30-year monthly mean temperatures (climatological mean) and their standard deviations (SDs) during March−May and September−November at Takayama-AMeDAS calculated at intervals of 10 years. In accordance with the Japanese Meteorological Agency, the climatological mean was defined as the average during 30 years at intervals of 10 years. We assumed that the year-to-year variability of monthly mean temperatures (i.e., the SDs) had an important effect on the modelled SGS and EGS dates.”

“Material and methods: 2.3. degree-day model of vegetation phenology” section (lines 198−206):

“First, we calculated the SGS dates from 1900 to 2020 at TKY-tower and the EGS dates from 1900 to 2019 at Takayama-AMeDAS. Second, we examined the long-term linear trends from 1900 to 2019/2020 and timeseries of 30-year monthly SDs, calculated at intervals of 10 years, of modelled SGS and EGS dates at TKY-tower, Takayama-AMeDAS, and their difference. The climatological mean of temperatures was defined as the 30-year average temperature at intervals of 10 years. Finally, to evaluate the effect of climate change, we examined the correlations between the SDs of 30-year monthly mean temperatures at timesteps of 10 years and the SDs of 30-year modelled SGS and EGS dates at TKY-tower, Takayama-AMeDAS, and their difference.”

“Results: 3.2. Air temperature” section (lines 291−297):

“Timeseries of 30-year monthly mean temperatures, calculated at intervals of 10 years (Fig. 4a, c), showed significantly positive rates of change in March, April, May, September, October, and November (Table 1).

Compared with May, the SD of the monthly mean temperature in March and April was always large (Fig. 4b). In contrast, compared with September and October, the SD of the 30-year monthly mean temperature in November was large from 1900−1929 to 1930−1959. In addition, the SDs have gradually increased since 1930 (Fig. 4d).”

Comment 5: 

In 3.4 – there should not be used the expression „validation“ – there are no data which were validated, there is only information about lets say comparison of modelled values with pictures from phenological cameras. If you want to do validation it is necessary to use some statistical indicators (e.g. RMSE) for calculating the terms of the start of growing season and end of GS. Only comparison with pictures is not enough for validation.

Answer: 

As you mentioned, our validations may not be sufficient. However, the modelled SGS and EGS dates may be affected by the spatial heterogeneity of tree species and their distribution, and microtopography. Therefore, we have changed the verb: “validate” into “assess”.

Comment 6:

In 3.4 – lines 258 and 261 – this is discussion not results.

Answer: 

Your assigned line numbers may be incorrect. Anyway, in relation to the answer for comment 10, this part is not discussion.

Comment 7: 

In 3.5 – first sentence – it is not clear for me, I cannot understand what the authors wanted to say. It should be maybe comparison of modelled data of start and end of GS with satellite data. Anyway the text is somehow heavy.

Answer: 

We have revised as follows (lines 326−343).

“Table 3 shows the relationships between the modelled SGS and EGS dates and the closest satellite-observed dates. Except for 2018, we could not obtain satellite data with less than or equal to 20% cloud coverage at about the times of the four modelled dates. The satellite-observed dates were 2 or 3 days later than the modelled SGS and EGS dates. On the satellite-observed date closest to the modelled SGS date at Takayama-AMeDAS (3 days later than the modelled SGS date), leaf flush had not occurred widely throughout the river basin (Fig. 7a). The GRVI of deciduous, broad-leaved forests was less than or equal to 0.0 (Fig. 7b). On the satellite-observed date closest to the modelled SGS at TKY-tower (2 days later than the modelled SGS date), leaf flush had expanded throughout the basin (Fig. 7c), and the GRVI in deciduous, broad-leaved forests had increased to 0.2–0.4. In addition, the GRVI around TKY-tower had increased to 0.1–0.4 (Fig. 7d). In contrast, on the satellite-observed date closest to the modelled EGS at TKY-tower (3 days later than the modelled EGS date), leaf colouring and leaf fall had occurred at elevations greater than 1000 m (Fig. 8a). The GRVI in deciduous, broad-leaved forests was less than or equal to 0.0 (Fig. 8b). On the satellite-observed date closest to the modelled EGS at Takayama-AMeDAS (2 days later than the modelled EGS date), leaf colouring and leaf fall had advanced throughout the basin (Fig. 8c), and the area characterized by GRVIs less than or equal to 0.0 had expanded in the basin (Fig. 8d).”

Comment 8: 

In 3.5 – I’m wondering if the satellite data should be used within this study at all. Information from only one year cannot say enough.

Answer: 

The modelled SGS and EGS dates may have been affected by the spatial heterogeneity and distribution of tree species as well as by microtopography. For this reason, analysis of SENTINEL-2A/2B satellite observations was a useful way to fill spatial gaps between modelled dates in a degree-day model of phenology and in situ observed data at a validation site [33]. Despite the short observation period of the SENTINEL-2A/2B satellites (since 2015 for SENTINEL-2A and 2017 for SENTINEL-2B), the SENTINEL-2A and 2B satellites are the only non-commercial satellites that have provided accurate geographical distributions of plant phenology in a steep river basin with a high spatiotemporal resolution.

We have added this in the “Material and method” section (lines 235−246).

Comment 9: Discussion

In 4.1 – first sentence – it is result not part for discussion. Lines 290-291 – not clear why authors compared their results with other species when we do not know which species were used by authors. Maybe the comparison with location or simply with other papers should be better.

Answer: 

We have moved this sentence into “Results” section (lines 277−280) and revised the first sentence of “Discussion” section as follows (lines 347−348). 

“The results of our analysis of the sensitivity of modelled SGS and EGS dates to elevation (Fig. 2) implied the following relationships. The modelled SGS date …”

As we mentioned the answer for comment 3, we have added the information of tree species.

Comment 10: 

Lines 311-312. If I understand well the authors said that SGS and EGS dates defined by satellite only in one year 2018 differed from the dates observed in-situ. I’m not sure if data from only one year are suitable for such conclusion.

Answer: 

As we answered for comment 8, this is one of limitations of our study. However, we should assess the representativeness of modelled SGS and EGS dates, which may be affected by the spatial heterogeneity of tree species and their distribution, and microtopography. Despite a short observation period of Sentinel-2A/2B satellites, we think that the Sentinel-2A/2B satellites-observed data are suitable to assess the geographical representativeness of modelled SGS and EGS dates.

We have calculated the correlation between the modelled SGS and EGS dates, and observed SGS and EGS dates shown in Table 1 by Nagai et al. (2013) from 2004 to 2011. Therefore, we have added the following descriptions in the “Results” section (lines 312–314 and 318−320). 

“In addition, the modelled SGS dates from 2004 to 2011 correlated significantly with the observed SGS dates shown in Table 1 of Nagai et al. (2013) (Spearman’s rank correlation coefficient, ρ = 0,94, p < 0.001) [32].”

“In addition, the modelled EGS dates from 2004 to 2011 correlated significantly with the observed EGS dates shown in Table 1 of Nagai et al. (2013) (Spearman’s rank correlation coefficient, ρ = 0.97, p < 0.001) [32].”

Comment 11: 

In 4.4 – lines 385-391 – this is rather methodology but absolutely not the discussion.

Answer: 

Your assigned line numbers may be incorrect. Therefore, we are very sorry but we have no idea about your comment.

Comment 12: 

4.4 – the importance of this chapter is unclear for me. Maybe it is important but there is no explanation by authors why this is important for the whole study. This should be excluded or rewrite.

Answer: 

At the beginning of this section, we have added the following explanation (lines 473−476). 

“We used phenology and live camera images taken in the upper and lower reaches of the Daihachiga River basin as validation data (Figs 5, 6). However, evaluation of the uncertainty of the spatial representation of phenological events by our model has been constrained by the amount of data available to us.”

Comment 12: Conclusion 

– I’m missing conclusions and information is authors fulfil and answer the goals of the study. I’m not sure what authors wanted to say by this study at all.

Answer:

We have revised as follows (lines 503−525).

“We used a degree-day phenology model to evaluate the spatiotemporal variability of the SGS and EGS dates in the Daihachiga River Basin on a century timescale and the relationship of that variability to changes of temperature. We found that (1) the sensitivity of the modelled SGS and EGS dates to elevation changed from 3.29 days (100 m)−1 (−5.48 days °C−1) and −2.89 days (100 m)−1 (4.81 days °C−1) , respectively, in 1900 to 2.85 days (100 m)−1 (−4.75 days °C−1) and −2.84 day (100 m)−1 (4.73 day °C−1), respectively, in 2019, and (2) the long-term trend of the sensitivity of the modelled SGS date to elevation was −0.0037 day year−1 per 100 m, but the analogous trend in the case of the modelled EGS date was not significant. Despite the use of daily phenology and live camera images as well as satellite data with a high spatial resolution to assess the generality and representativeness of the modelled SGS and EGS dates, limitations of our study still remain. The next steps will require (1) examination of the relationship between elevation and SGS and EGS dates in other steep river basins where annual mean temperatures are much higher than the temperatures in the Daihachiga River basin, (2) integrative evaluation of the SGS and EGS dates based on a degree-day phenological model and satellite observations with a fine spatial resolution and high observation frequency, (3) acquisition of an accurate basin-specific temperature lapse rate in each target basin, and (4) further acquisition of ground truth data such as live camera images and images with geotags on the Internet. Plant phenology was more sensitive to elevation than to latitude in a local area (e.g., within 2° latitude × 3° longitude) [9], where we could ignore the effect of phenological plasticity associated with climate change. Hence, the development of degree-day phenological models in multiple steep river basins with different climate conditions will deepen our ecological understanding of the sensitivity of spring and autumn phenology to future climate change.”

Reviewer 2: 

The manuscript entitled ‘How did the characteristics of the growing season change during the past 100 years at a steep river basin in Japan?’ by Nagai et al deals with the growing season in a mountainous region of Japan. Paper is scientific but needs improvement. My observations on the follow up correspondence are as follows:

Answer: 

Thank you for your efforts to improve our paper. In accordance with your kind and constructive comments, we have revised the manuscript.

Comment 1: Does number of sunny days play any role in the higher elevation to the phenology? I do agree with the author about the urban heat island explanation but want to see if temperature gradient with available sunny days have any impact needs to be studied. 

Answer:

In Japan, the timing of leaf flush (i.e., SGS date), and leaf colouring and leaf fall (i.e., EGS date) in Japan was mainly explained by temperatures. Therefore, we have added this in the “Material and methods” section as follows (lines 179−180).

“The timing of leaf flush (i.e., SGS date) as well as leaf colouring and leaf fall (i.e., EGS date) in Japan are explained mainly by temperatures [8, 9, 53, 54].”

Comment 2: Use of the punctuation marks and conjunction e.g. ‘and’ in the abstract and introduction section need revision [e.g. L-27, 68-69] due to unnecessary use at times. 

Answer: 

We have revised lines 31 and 71.

Comment 3: The North arrow is an important component of the map and therefore it must be incorporated in each map.

Answer: 

We have shown the North arrow in each map (please see Figs. 1, 7, and 8).

Comment 4: What are the limitations of the study? Elaboration a little on this aspect of the study. 

Answer: 

As we mentioned “Conclusion” section, we have still remained the limitations of our study for assesses of the generality and representativeness of modelled SGS and EGS dates. To clarify these limitations, we have revised “Discussion” (lines 474−476) and “Conclusion” sections (lines 503−513). Please see the answer for comment 10. 

Comment 5: L- 210 should be “We used RGB composite images and calculated the green-red vegetation index (GRVI; [54]).” 

Answer: 

We have revised lines 250−251.

Comment 6: Mention in the article, if any cloud removal method used in the research study to get accurate information from the SENTINEL data. 

Answer: 

We have added the following information (lines 252−256).

“In this study, atmospheric corrections were performed by using the SNAP (Sentinel Application Platform) toolboxes (http://step.esa.int/main/toolboxes/snap/, accessed 6 July 2021). We did not apply further cloud masking to the satellite data because there was no cloud contamination in our target river basin.”

Comment 7: Used acronyms need to be in their full form the first time they are used. This issue is consistent in the whole manuscript. 

Answer: 

We have shown the full form of acronyms (lines 79, 81, 402, and 492). 

Comment 8: The method used to calculate the trend, correlation result, and significance of the results need to be elaborated. The level of significance used to consider the result ‘significant’ should also be mentioned. 

Answer: 

In Fig. 2, we have added the 95% confidence interval of each linear trend as follows. 

Fig 2. Time-series of modelled (a) SGS and (b) EGS dates at TKY-tower (1420 m a.s.l.) and Takayama-AMeDAS (560 m a.s.l.). (c, d) Differences between sites in modelled (c) SGS and (d) EGS dates (Takayama-AMeDAS dates minus TKY-tower dates). The dashed lines show statistically significant (p < 0.05) linear trends and their 95% confidence intervals.

Comment 9: Camera images and satellite observation were used to validate the obtained results. Therefore, it is better to make one common section 3.4 for validation through camera image and satellite observation. Otherwise, section 3.4 can be split into two sub-sections i.e. one for camera images and the other one for satellite observation. 

Answer: 

We have made one common section and revised the title of sub section.

Comment 10: The conclusions section is very weak. It should provide a summary of the whole study, including major results of the study, method, issue addressed. 

Answer: 

We have revised as follows (lines 503−525).

“We used a degree-day phenology model to evaluate the spatiotemporal variability of the SGS and EGS dates in the Daihachiga River Basin on a century timescale and the relationship of that variability to changes of temperature. We found that (1) the sensitivity of the modelled SGS and EGS dates to elevation changed from 3.29 days (100 m)−1 (−5.48 days °C−1) and −2.89 days (100 m)−1 (4.81 days °C−1) , respectively, in 1900 to 2.85 days (100 m)−1 (−4.75 days °C−1) and −2.84 day (100 m)−1 (4.73 day °C−1), respectively, in 2019, and (2) the long-term trend of the sensitivity of the modelled SGS date to elevation was −0.0037 day year−1 per 100 m, but the analogous trend in the case of the modelled EGS date was not significant. Despite the use of daily phenology and live camera images as well as satellite data with a high spatial resolution to assess the generality and representativeness of the modelled SGS and EGS dates, limitations of our study still remain. The next steps will require (1) examination of the relationship between elevation and SGS and EGS dates in other steep river basins where annual mean temperatures are much higher than the temperatures in the Daihachiga River basin, (2) integrative evaluation of the SGS and EGS dates based on a degree-day phenological model and satellite observations with a fine spatial resolution and high observation frequency, (3) acquisition of an accurate basin-specific temperature lapse rate in each target basin, and (4) further acquisition of ground truth data such as live camera images and images with geotags on the Internet. Plant phenology was more sensitive to elevation than to latitude in a local area (e.g., within 2° latitude × 3° longitude) [9], where we could ignore the effect of phenological plasticity associated with climate change. Hence, the development of degree-day phenological models in multiple steep river basins with different climate conditions will deepen our ecological understanding of the sensitivity of spring and autumn phenology to future climate change.”

---

## [Editor Report · Decision Letter 2]

12 Jul 2021

How did the characteristics of the growing season change during the past 100 years at a steep river basin in Japan?

PONE-D-20-20028R2

Dear Dr. Nagai,

We’re pleased to inform you that your manuscript has been judged scientifically suitable for publication and will be formally accepted for publication once it meets all outstanding technical requirements.

Kind regards,

Dusan Gomory

Academic Editor

PLOS ONE
---

## [Editor Report · Acceptance letter]

21 Jul 2021

PONE-D-20-20028R2 

How did the characteristics of the growing season change during the past 100 years at a steep river basin in Japan? 

Dear Dr. Shin:

I'm pleased to inform you that your manuscript has been deemed suitable for publication in PLOS ONE. Congratulations! Your manuscript is now with our production department. 

Kind regards, 

on behalf of

Dr Dusan Gomory 

Academic Editor

PLOS ONE